# The Spatial Pedigree in Traditional Villages under the Perspective of Urban Regeneration—Taking 728 Villages in Jiangnan Region, China as Cases

**Xinqu Liu** [1,*], **Yaowu Li** [2], **Yongfa Wu** [3] **and Chaoran Li** [1]

1   School of Architecture, Yantai University, Yantai 264005, China
2   School of Architecture, Tsinghua University, Beijing 100190, China
3   Gold Mantis School of Architecture, Soochow University, Suzhou 215006, China
*   Correspondence: liuxinqu@ytu.edu.cn

**Abstract:** In current urban regeneration, the conservation and reuse of traditional village space are significant measures to activate urban-rural development. Traditional villages in the Jiangnan region of eastern China represent the typical vernacular culture of urban-rural settlements, which affects the dynamic development of urban regeneration. Aiming at the disadvantages of the decentralization of research objects and the simplification of spatial characteristics in the research of traditional villages in the Jiangnan region, this paper will construct the spatial pedigrees of villages. This study takes 728 traditional villages in the Jiangnan region as cases. First, through on-site research, the spatial pattern of villages in isolated areas was mapped and drawn. Then, on the basis of on-site review, this study labeled the village space, summarized village space information, and used ArcGIS and BIM to create a 3D model of the village. In ArcGIS platform through SOA to establish criterion framework for recognizing the types of village pedigree. Finally, the classification of villages was spatially visualized, and a pedigree was constructed according to the type context. The research result indicates: (1) The aggregation mode of traditional villages in the south of the Jiangnan region presents the characteristics of local aggregation and partial dispersion, which are directly affected by factors such as elevation, water system, and road system. (2) Analysis of the spatial pedigrees of traditional villages through spatial types and spatial growth patterns revealed four pedigree types, including spatial environment pedigree, the spatial organization pedigree, spatial morphological pedigree, and architectural form pedigree. (3) The pedigree shows and distinguishes independent developmental context and evolved form, presenting an extended pattern of prototypes and sub-types.

**Keywords:** traditional village; village space; pedigree; Jiangnan region; urban regeneration

## 1. Introduction

### 1.1. Traditional Village Protection and Urban Regeneration

The protection and reuse of traditional villages are important topics for urban regeneration. Its purpose is to activate the village space through reasonable methods, and to revitalize urban space which is facing decline [1]. Protection of Traditional Chinese villages is based on the perspective of urban development, the organic renewal of space, and paying attention to the comprehensive influence of social, economic, artistic, cultural, and other factors [2]. Traditional villages and cities, as part of China's human settlement space, are an active element and a restriction in the constitution of the environment. Since 2012, China has carried out the investigation of traditional villages. In 2017, rural revitalization became a significant tactic focusing on the coordinated development between urban and rural areas and exploring the sustainable development mode of traditional villages from the perspective of ecological balance. Meanwhile, with the influence of rapid urbanization, the spatial form and survival mode of traditional villages in China has changed from the self-organizing to the other-organizing [3,4]. The decline of traditional culture and

the collapse of local power led to the need to explore a new transformation model for traditional village space. A phenomenon of identical boring facades for different urban spaces is becoming an essential negative factor for the urban-village human environment. The following three facts are responsible for this situation [5–7]:

(1)　The traditional culture in urban areas like traditional villages or heritage spaces is not thoroughly understood and, therefore, insufficiently appreciated and represented in urban design, which lacks basic village spatial information.

(2)　Undifferentiated classification of traditional village spaces due to a lack of basic information, along with the importance of cultural conservation diversity in urban regeneration being ignored, has led to the loss of urban-village characteristics.

(3)　The similar reuse conservation method being used for urban regeneration in different urban-village spaces has weakened the urban cultural heritage and increases the construction consumption.

Therefore, it is necessary to sort out the diverse characteristics of traditional village space, and to clarify its similarity and individuality with the spatial type context, that is, the spatial pedigree. Exploring a traditional pedigree to excavate and analyze the key of traditional culture through prototype and variation of the spatial form is like the gene in biology [8–10]. Diverse village characteristics and spatial forms which present in the pedigree can provide inspirations for spatial design and for the protection and reuse method of urban regeneration [11–13].

### 1.2. Conservation and Regeneration of Traditional Villages in Jiangnan Region, China

The Jiangnan region (Figure 1), located in the Yangtze River Delta in China, has a unique historical heritage due to its geographical advantages, it is close to water and mountains and has a humid climate, as well as due to the long-term integration of North and South cultures [14]. As a heritage of farming civilization in the Jiangnan region [15], traditional villages are part of contemporary China's regional cultural development and urban regeneration, and have a synergistic role with the developing eastern coastal urban space. Historically, the Jiangnan region has been in a multi-integrated area [16]. The impact of culture, history, folk customs, and other factors has made the village space present a variety of spatial forms and local ecological wisdom. Up to now, China has completed five evaluations of traditional villages. There are 6819 traditional villages in the country [17]. Among them, there are 728 traditional villages in the Jiangnan region of this paper. In 2019, the list of traditional villages in the Jiangnan region was announced. The large number and wide distribution involved increased the difficulty to research or conserve them [18]. Therefore, it is necessary to integrate the scattered resources, excavate the characteristic features of the space, sort out the internal logic of its change mechanisms and influences, and provide the basic materials for urban regeneration. The key to active protection in urban regeneration relies on the protection of style and features guided by local characteristics, and the extraction of spatial symbols based on a cultural core for transformation and reuse [19].

For instance, there is a representative reuse case in urban regeneration in Hangzhou City, the Qin Chuan village (Table 1). In this case, designers focus on the village space characteristics so that well-maintained natural landscapes and crops can provide livelihoods through in-site conservation methods in urban regeneration. Conserved and repaired, the historical architectures of the village are a reuse material model for cultural inheritance, and provide space for tourism, cultural learning, etc. In order to activate the traditional villages in the Jiangnan region, the visualization of village big data and the construction of pedigrees are urgent problems to be solved. It is necessary to supplement the basic information of the village through on-site investigation and research, and to complete the contextual sorting on the basis of regional elements. In order to activate the traditional villages in the Jiangnan region, the visualization of traditional villages' big data and construction of their pedigrees are urgent purposes.

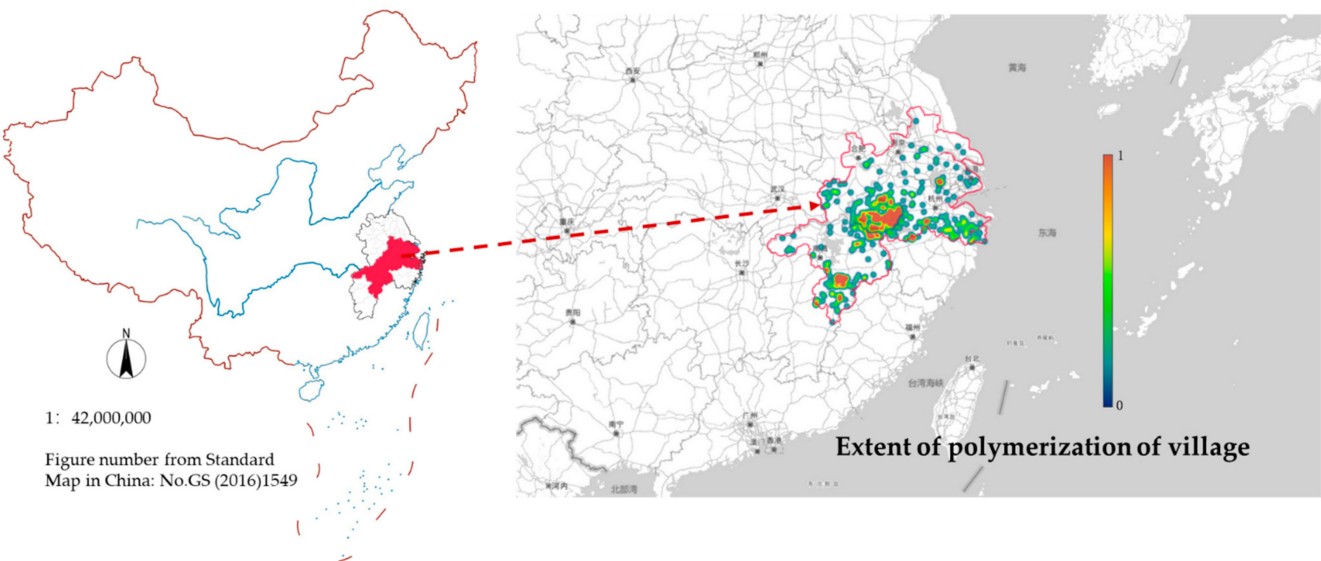

**Figure 1.** Jiangnan region of China: edited by authors.

**Table 1.** Urban regeneration typical sample of traditional villages in Jiangnan region: edited by authors, information is obtained from field research, photographs taken by authors.

| Qin Chuan Village in Hangzhou City | Landscape | Road System and Architectures | Ancestral Halll |
|---|---|---|---|
| 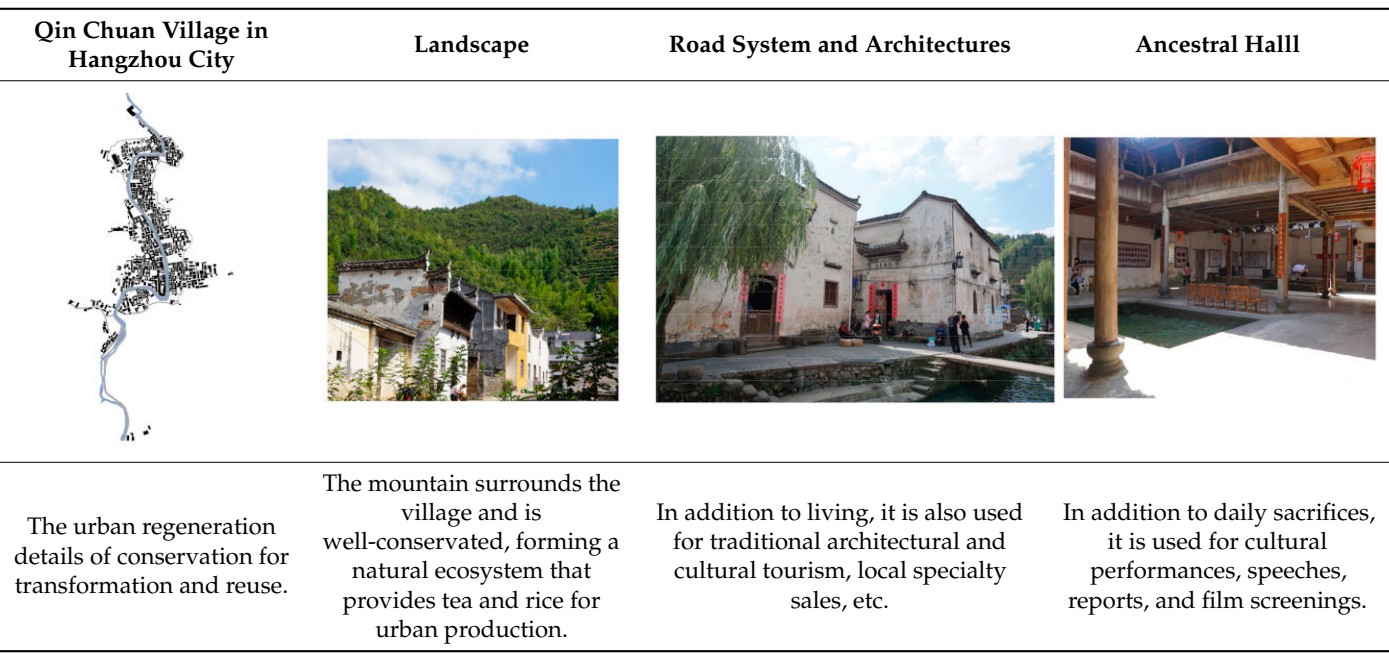 | | | |
| The urban regeneration details of conservation for transformation and reuse. | The mountain surrounds the village and is well-conserved, forming a natural ecosystem that provides tea and rice for urban production. | In addition to living, it is also used for traditional architectural and cultural tourism, local specialty sales, etc. | In addition to daily sacrifices, it is used for cultural performances, speeches, reports, and film screenings. |

### 1.3. Literature Review

1.3.1. Pedigree Construction of Traditional Village Space

Pedigree, also known as genealogy, is a concept that originates from anthropology and philosophy. The pedigree at the anthropological level mainly includes three levels of content [20]: (1) books describing the lineage or the historical system of similar things; (2) a family system; (3) a system of species changes. Pedigree in anthropology emphasizes the systematic nature of things and species. At the philosophical level, the concept of genealogy comes from Nietzsche's The Genealogy of Morals [21], and it has been further developed in Foucault's philosophical theoretical system [22]. Genealogy is an analytical method in philosophy, as well as a deeply self-critical philosophical point [23]. In this study, in terms of the material characteristics of traditional village space, its attributes require it to be defined as pedigree. In early research on traditional villages, such as that of the

American scholar Rudowski, there was an architectural analysis of the vernacular pedigree space [24]. Likewise, scholars such as the Frenchman Pierre Lamay used history as a clue to connect the development pedigrees of vernacular architecture in traditional villages [25]. In recent years, scholars such as Whitehand have focused on the development of China's traditional village areas, and have pointed out that the method of urban-village synergy can be explored through the pedigree of spatial forms [26]. However, in existing relevant literature on the pedigrees of traditional villages, scholars have mostly focused on record-keeping of the historical context, and lack systematic construction of the village space. With the advancement of sustainable research on traditional villages in China in recent years, scholars have focused on the necessity and application of traditional village pedigree establishment. Scholars such as Chang Qing and Luo Deyin proposed that construction of the traditional pedigrees of vernacular architectures or villages in China is an operational method for environmental protection and sustainability of contemporary regional terroir, which is significant to explore the integration of future buildings from the perspective of urban regeneration [11,12].

The concept of traditional village pedigrees is derived from the concept of genealogy in anthropology and philosophy, focusing on the context of the development of things, while including the critical philosophical thinking. From the perspective of research methods, the contextualized expression of traditional village pedigrees has absorbed the systematic methods of anthropological pedigrees [27]. There are three methods to research traditional village pedigrees: dynamic traceability history (DTH) [28,29], static analysis type (SAT) [30,31], and dynamic and static analysis of context (DAA) [30–33]. DTH focuses on the dynamic line of village development [34], while SAT focuses on building types and characteristics at the same time. Both methods lack the comparison and analysis of the changing relationship between villages [35]. The DAA is an indispensable method for contemporary traditional village pedigree research. The advantage of DAA is that it can connect the extravagant and scattered information systems in series. However, it needs an amount of data to expand the sample for comparative analysis, which is necessary for integrating it into contemporary visualization methods [36]. Therefore, this paper uses Spatial Overlay Analysis (SOA) as a method, taking 728 traditional villages in the Jiangnan region as cases to expand the typical sample.

### 1.3.2. Research on Village Spatial Visualization Using GIS and BIM as Tools

Since the concept of GIS was proposed by Canadian surveyor Roger F. Tomlinson in 1963, the field of the traditional village has been used in early analysis in the last century [37]. In recent years, it has been applied to the spatial distribution or environmental landscape analysis of contemporary villages [38,39]. Represented by scholars such as Nieto Masot, C. Conrad, and R.J. Hewitt, the characteristics of ArcGIS engineering data management and planning simulation are being used to analyze population, land status, economic development, natural landforms [40,41], etc. The data management and model analysis in ArcGIS provide a visual and effective approach to village conservation [42]. These studies have focused on the exploration of macro-influencing factors such as village distribution and economy, the research areas are limited to individual cases, and they lack systematic research on the spatial forms of villages [43]. The application of tools such as ArcGIS and BIM (Building Information Modeling, which includes ArchiCAD, Photoshop, Adobe Illustrator, etc.) in Chinese traditional villages conservation has became a method to explore urban regeneration area in recent years. It has been widely used in analyzing the style and characteristics of villages, which have positive significance for the conservation and utilization of contemporary villages [44,45]. It includes the spatial macro-distribution characteristics of Chinese villages, and the method of village conservation based on ArcGIS data processing [46]. The use of ArcGIS and BIM applied to Chinese traditional villages is reflected in the following three aspects: (1) organizing the village space resources and providing a digital research path [47]; (2) evaluating villages through macroscopic spatial analysis [48]; and (3) scientifically quantifying the macro data of the

village's environmental landscape, population distribution, etc., and providing a reference for the village planning and design [49]. However, at this stage, the use of ArcGIS and BIM in the villages lacks the analysis of the spatial form on two levels: (1) the meso-level, including village landscape, road system, river system, boundary outline, etc.; and (2) the micro-level, including architecture outline, public space elements, architecture forms, etc. Although there are some researchers who have used machine learning techniques to recognize the rural area, it has been limited to identifying the difference between the rural and urban or the boundary of a village [50]. The HR-RSF-UV framework can accurately identify the boundaries of urban villages (UVs), but still fails to provide identification methods for features such as streets and architectures inside the villages. It also ignores the cross-correlation between multi-source spatial features [51]. Meanwhile, the pix2pix model has been used to repeatedly label the spatial pattern, but the research space needs to be based on manual labels, and the targeted spaces are small-scale manors [52]. Therefore, in order to accurately identify the village space, and to sort out the cross-correlation between multiple villages, this paper will construct spatial pedigrees, which clarify the prototype and sub-type form (Table 2). In conclusion, this paper takes advantage of the spatial information management advantages of the ArcGIS and the accuracy of BIM modeling to label the model space and build a 3D model on the basis of the village space surveyed and mapped onsite. Moreover, according to the classification criteria, villages were filtered and located on the ArcGIS platform, and the spatial pedigree of the villages was established by the visualization of the type context; the spatial pedigree can then be used to explore the characteristics of village space and the possibility of historical conservation in contemporary urban regeneration.

**Table 2.** The different methods to recognize villages: edited by authors.

| Method | Dual-Branch Deep Neural Network | HR-RSF-UV | Pix2Pix Model | On-Site Surveying and Modeling and Labeling in ArchiCAD |
|---|---|---|---|---|
| |  |  |  |  |
| Features | Model yields a high accuracy of 92.61% in Jing-Jin-Ji region of China. Limited to identifying the difference between urban village (UV) and non-urban village (non-UV); not suitable for isolated and remote areas, such as traditional village. | Can accurately identify the boundaries of urban villages. Fails to provide identification methods for features such as streets and architectures inside the villages; ignores the cross-correlation between multi-source spatial features | Pix2Pix model can generate private garden layout by inputting site plan with certain conditions. Simplified the complex spatial form and is not suitable for larger-scale villages. | Providing 1:1 scale and labeling of village, architecture, landscape, and roads one can recognize clearly. Can input and accurately identify the village space in ArcGIS, and sort out the cross-correlation between multiple villages. |

In this research, we propose a criterion framework to identify village types and construct village pedigrees. First, based on a large number of on-site investigations, the lacking village information for isolated areas is composed. Second, combining the ArcGIS data management ability and the BIM accurate modeling ability, the software builds the 3D village model and forms the village database. Through ArcGIS software, the characteristics of village space can be analyzed at the macro, meso, and micro levels. The influence factors of the village are obtained by analysis at the macro level, and the classified criteria to recognize the traditional village spatial pedigree can be obtained by SOA analysis at the

meso and micro levels. This pedigree has the ability to identify and classify villages, and can be used to analyze village cross-correlation and types. The main contributions of this paper are as following:

(1) Obtained the village spatial information in the isolate areas of the Jiangnan region, and established a village database, which used for the digital management of village conservation.

(2) Established the model working path between ArcGIS and BIM, completed built 3D village model though 2D spatial information, which can be used for the visualization of village protection;

(3) Established criterion framework can be command in ArcGIS to identify village spatial classifications and construct pedigree.

(4) Village pedigree presents the types and the cross-correlation between multiple villages, and the similarities and differences of village types can be inspired for urban-village conservation and reuse.

## 2. Materials

### 2.1. Research Area

The research area of this paper focuses on Jiangnan, China, including five areas: Shanghai, northern Jiangsu, southern Anhui, northeastern Zhejiang, and northeastern Jiangxi, with a total of 728 traditional villages. In terms of overall quantity, the distribution of traditional villages in the south of the Jiangnan region is unbalanced. It shows wide distribution, large geographical scope, and uneven number (Figure 2). The time span of the villages includes traditional villages from the Ming and Qing Dynasties up to the present, with a few dating back to the Tang and Song Dynasties [53]. The water resources in the Jiangnan region include lakes, rivers, and oceans, and with the changeable mountainous terrain, traditional village space needs to adapt to the natural environment. Therefore, traditional villages in the Jiangnan region show distinguished craftsmanship and diverse spatial forms [54]. As a research sample of traditional villages, it has the cultural representation and spatial characteristics showing the unique context of the pedigree [55].

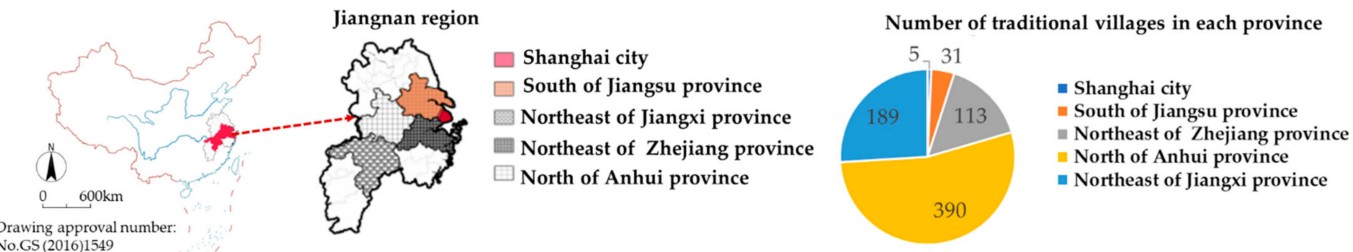

**Figure 2.** Research area and the number of villages: edited by authors.

### 2.2. Data Sources

The data for this study come from three sources, including official websites and institutions, local book resources, and field research:

(1) Village name, address, and basic information data for this paper come from the Ministry of Housing and Urban-Rural Development of the People's Republic of China and the Research Center for the Protection and Development of Chinese Traditional Villages [56,57]. The satellite map and map data are from the BIGEMAP high-definition satellite map resource library. The scale of the illustration is 1:1, which can clearly identify the scale, outline, and location of the village, and can be adjusted to match the pedigree.

(2) Qualitative data such as the history, culture, social form, and ethnic customs of the village are derived from the village annals, local chronicles, and other village

books, and are obtained through field investigations. The time span is from the Tang Dynasties to 2022.

(3)  The physical space forms such as village outline boundary, village roads, architectural style, etc., are obtained by on-site surveying, survey and draw, and the scale is 1:1. The scale of the village picture can be adjusted according to the situation. Other materials that cannot be obtained directly from books or official websites can be obtained through field research or interviews with villagers, such as local legends, population changes, etc.

## 3. Method

In this research we construct village spatial pedigrees to analyze the relationship of traditional villages in the Jiangnan region, which includes three major steps (Figure 3). First, based on the on-site investigation to obtain the 728 villages' information data and village spatial data, ArchiCAD is used to label village space and build a village 2D model. Inputting these village data to ArcGIS, a database is formed on this platform. Second, using Nearest Neighbor Index (NNI) to analyze the aggregation patterns of the 728 villages, we obtain three main factors which influence traditional villages in the Jiangnan region. Dividing these three macro factors from the meso and micro levels in the ArcGIS platform by Spatial Overlay Analysis (SOA), we obtain the criteria for recognizing the types of village spatial pedigree. Finally, inputting the framework to ArcGIS, we use command filtering to classify the prototypes and sub-types to construct the pedigree, and we open a village space information table in ArcGIS to locate representative villages and select the model image to visualize the pedigree (Figure 3).

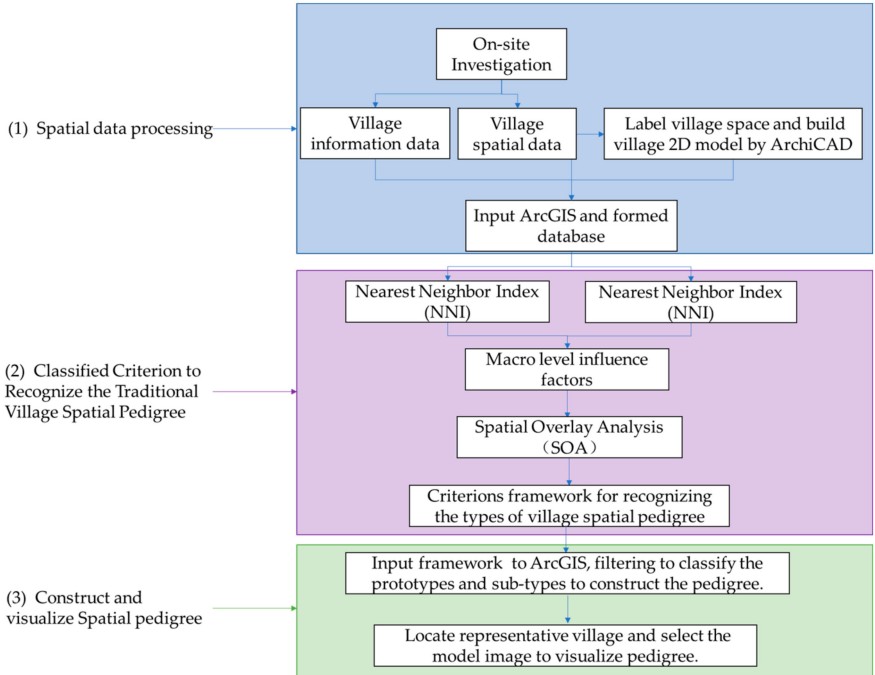

**Figure 3.** Research steps: edited by authors.

### 3.1. Spatial Data Processing

3.1.1. On-Site Investigation

On-site investigation, also called field research, is a practical method of village research, which obtains the real data of the village intuitively [58]. This study obtains visual spatial image data through aerial photography of drones, obtains village site data through surveying and draw instruments, and understands the current situation of the village through on-site interviews. Jiangnan traditional villages have minimal traffic accessibility due to the limitation of topography, landform, and climate conditions, and it is difficult

to obtain spatial data by official internet or public websites like Baidu or Google or other documentary archives directly. Authorized organization and other materials are based primarily on location, name, or historical information. In addition, for villages influenced by urbanization, the situation of villages is unpredictable compared with the past, so that spatial information materials need to be obtained directly through on-site investigation. This paper conducted on-site investigation for traditional villages in the Jiangnan region that lacked graphic and data information, obtained direct first-hand information, and completed village spatial data by survey and draw on-site.

### 3.1.2. Label Village Space and Build Village Model

In this step, using ArchiCAD to draw and review the village spatial 2D model, and collecting the data of the village space, the scale is 1:1(Appendix A, Table A1). ArchiCAD is used to label village spatial layers, including the traditional architecture, river system, landscape or mountain system, and road system (Table 3). These village space layers are input to ArcGIS, and form the village information database, which includes basic data and spatial information for each village. Every village will have a spatial information table in the ArcGIS platform. Global mapper transforms DEM data to UTM image, rasterizes the 2D village image, and puts it into ArcScene, then overlays the DEM, boundary and other spatial information to generate the 3D model (Figure 4).

**Table 3.** Sample of survey and draw for traditional villages in Jiangnan region: edited by authors.

| Measures | Village Model | Outline Boundary Layer | Road System Layer | River System Layer |
|---|---|---|---|---|
| Obtained description | 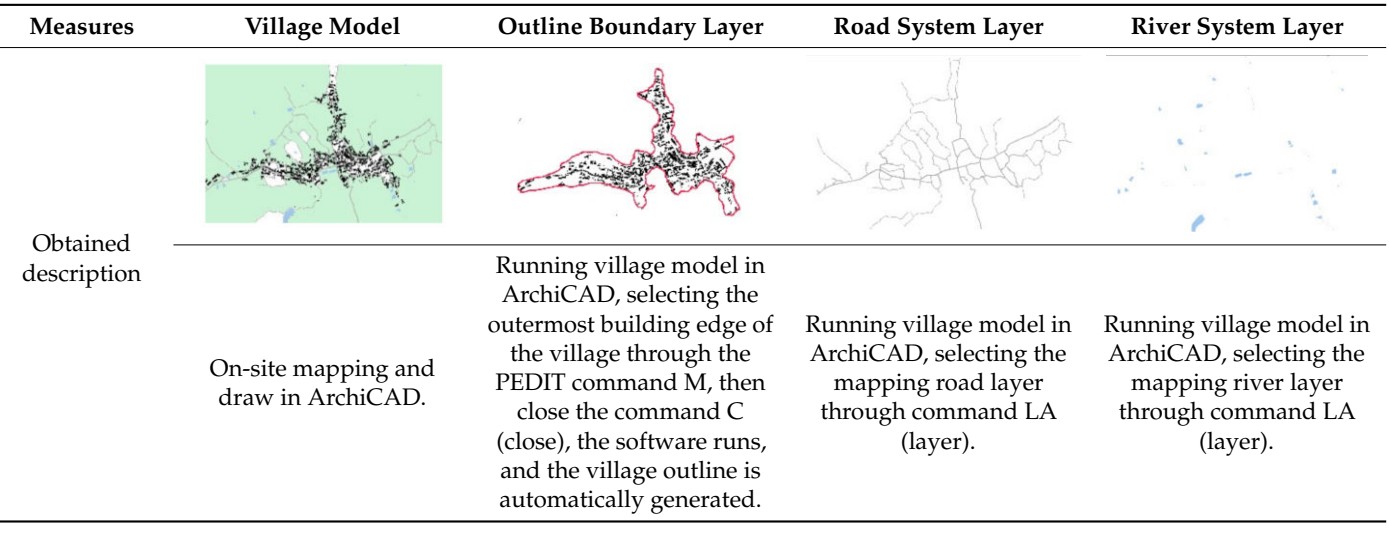 On-site mapping and draw in ArchiCAD. | Running village model in ArchiCAD, selecting the outermost building edge of the village through the PEDIT command M, then close the command C (close), the software runs, and the village outline is automatically generated. | Running village model in ArchiCAD, selecting the mapping road layer through command LA (layer). | Running village model in ArchiCAD, selecting the mapping river layer through command LA (layer). |

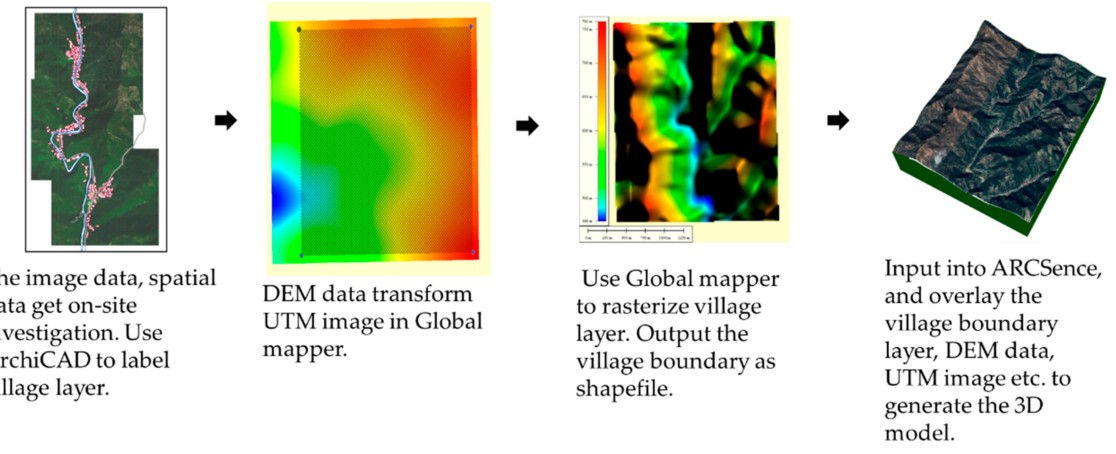

The image data, spatial data get on-site investigation. Use ArchiCAD to label village layer.

DEM data transform UTM image in Global mapper.

Use Global mapper to rasterize village layer. Output the village boundary as shapefile.

Input into ARCSence, and overlay the village boundary layer, DEM data, UTM image etc. to generate the 3D model.

**Figure 4.** The process to generate village 3D model: edited by authors.

*3.2. Classified Criterion to Recognize the Traditional Village Spatial Pedigree*

3.2.1. Nearest Neighbor Index (NNI)

NNI uses the distribution of random patterns as a standard to measure the spatial distribution of point elements. In the model of the village space, the data are abstracted into points for analysis [59]. We calculate the nearest neighbor distance of each point feature in the study area and take the average value, that is, the nearest neighbor distance of point features, denoted by d. The average value of the nearest neighbor distance of the random pattern of point elements is the theoretical nearest neighbor distance, which is represented by $d_{min}$. In the Complete Spatial Randomness (CSR), the average NNI can also be obtained, and its expectation is $\sum d_{min}$, represented by the following formula:

$$d_{min} = \frac{1}{n} \sum_{1=1}^{n} d_{\min} \tag{1}$$

In terms of the study, the average NNI in random mode is related to the area *A* of the study area and the number of events *n*, considering the boundary correction of the study area. This is represented by the following formula:

$$\sum d_{min} = \frac{1}{2}\sqrt{\frac{A}{N}} + \left(0.051 + \frac{0.041}{\sqrt{n}}\right)\frac{p}{n} \tag{2}$$

There are three types of spatial distribution of point elements in nature and society: random, uniform, and agglomeration. This paper uses the NNI to determine the spatial agglomeration type of traditional villages in the Jiangnan region. According to the average distance and the expected average nearest neighbor distance, the NNI will be calculated. If NNI = 1, it indicates that the observed event process comes from a random pattern, and the distribution pattern tends to be random. If NNI < 1, it indicates that many of the observed data points are relatively similar in space, and the distribution patterns tend to be clustered. If NNI > 1, it indicates that the nearest neighbor distance of the observed data points is greater than the nearest neighbor distance of the random pattern, and the distribution pattern tends to be scattered.

3.2.2. Kernel Density Estimation (KDE)

The KDE visually presents the density distribution of the village and is used to identify the scattered area and concentrated area of the village gathering. It reveals the changing process of village spatial density and can find regional differences in the overall pattern of village evolution [60]. This paper uses KDE to focus on expressing the spatial agglomeration characteristics of traditional villages in the Jiangnan region, and based on this, it extracts the spatial agglomeration pattern of rural settlements. The predicted density for the new (x, y) location is determined by:

$$Density = \frac{1}{(radius)^2} \sum_{i=1}^{n} \left[ \frac{3}{\pi} pop_i \left(1 - \left(\frac{dist_i}{radius}\right)^2\right)^2 \right]$$

$$for\ dist_i < radius$$

In the formula:

$i = 1, \ldots, n$ is the input point. Only include points in the sum if they are within a radius distance of the (x, y) location.

$pop_i$ is the population field value of the I point, and it is an optional parameter.

$dist_i$ is the distance between point *i* and the (x, y) location.

Then multiply the calculated density by the number of points, or the sum of the population field (if any). This correction takes such space quota equal to the number of points (or the sum of the population field) instead of equal to 1. A separate formula needs

to be calculated for each location where the density is supposed to be estimated. Since the raster is being created, the calculation will be implemented to the center of each element in the output raster.

### 3.2.3. Classified Criterion by Spatial Overlay Analysis (SOA)

The location, environment, pattern, form, and other data of village space are superimposed on the ArcGIS platform to generate a new data layer. The result combines the attributes of the original two or more layers of elements. SOA includes the comparison of spatial relationships and the comparison of attribute relationships. SOA can be divided into visual information overlay, point and polygon overlay, line and polygon overlay, polygon overlay, raster layer overlay, etc. Due to the software characteristics of ArcGIS, in the macro level at Jiangnan region scale, a village is displayed as a point; in the meso level at village scale, the village shows as a spatial layer; in the micro level at architectural scale, the village is a larger area than the architecture. From these three levels in ArcGIS, we can have the spatial layer relate to the level which includes the village data to identify the type of village (Table 4).

**Table 4.** The different village spatial data obtained from different levels in ArcGIS: edited by authors.

| Macro Level | Meso Level | Micro Level |
| --- | --- | --- |
| 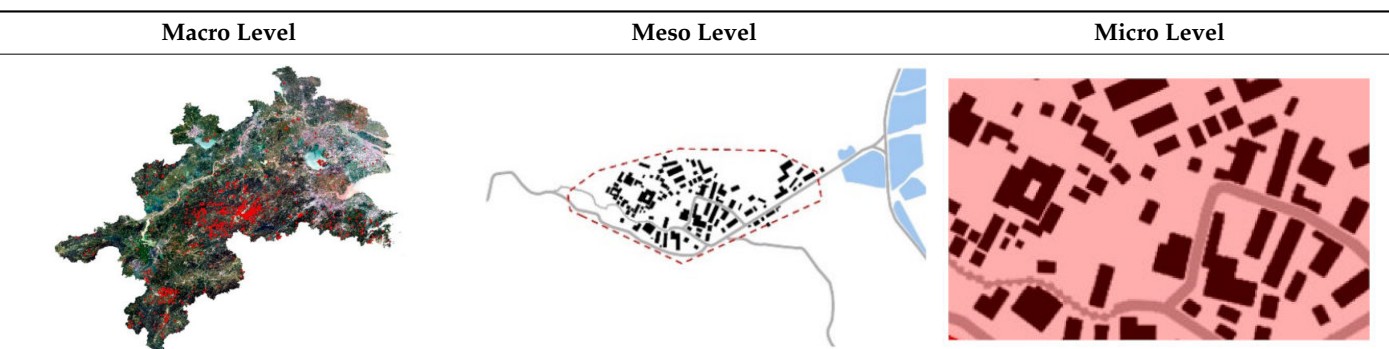 | | |
| Village represented by a point, including the village's location, name, distribution data. | Village represented by a spatial layer, including the village's outline boundary, river system, road system data. | Village represents a larger area than architecture, including the village's public space elements, etc. |

The three levels in ArcGIS are used, first, through NNI and KED to analyze the distribution and influencing factors of villages by point element in macro level, where the spatial aggregation pattern of traditional villages in the Jiangnan region is influenced by elevation, water system, and roads. Then, according to these three main influencing factors, we divided the spatial data by elevation, water system, and road system of the survey from meso level and micro level in ArcGIS by SOA. The rest of the spatial information we classified according to the other qualitative factors. Therefore, through the different level clues we obtain the criteria for recognizing the types of village spatial pedigree (Table 5, Appendix A, Tables A2–A5).

### 3.3. Construct and Visualize Spatial Pedigree

There are four criteria for identifying the types of village spatial pedigree. We use this criterion framework input to ArcGIS which includes 728 traditional villages in the Jiangnan region, and it will be formed as an attribute list in this engineering software database. Through the different criteria, utilizing the filter command to classify the prototypes and sub-types, the link between prototypes and sub-types constructs the pedigree. Then we choose the rest of the village elements in the ArcGIS software screen, to open the village space information table, locating this type of village accurately. In the previous method, we have obtained model images of villages, locating representative village prototypes and sub-types after filtering, and we select the model image into the pedigree process to form a visual pedigree (Figure 5).

**Table 5.** The criteria for identifying the types of village spatial pedigree: edited by authors.

| Level | Macro Level | Meso Level | | | Micro Level |
|---|---|---|---|---|---|
| | | Pedigree | | | |
| Research Object | Influencing Factors | Traditional Village Spatial Environment Pedigree | Traditional Village Spatial Organization Pedigree | Traditional Village Spatial Morphological Pedigree | Traditional Village Architectural Form Pedigree |
| Criterion 1 | Elevation | Terrain elements | Village architecture group | Village outline boundary style | Area |
| | | | | Number of village space unit | |
| Criterion 2 | Water system | Water elements | Water system style | Water system style | relationship with water |
| Criterion 3 | Road system | Road elements | Village road style | Village road style | relationship with road |
| Criterion X (X ≥ 3) | Other qualitative factors | ——— | Public space elements are used for village's daily activities and gathering centers | Feng shui planning principles, local art worship, etc. | Architecture elements |

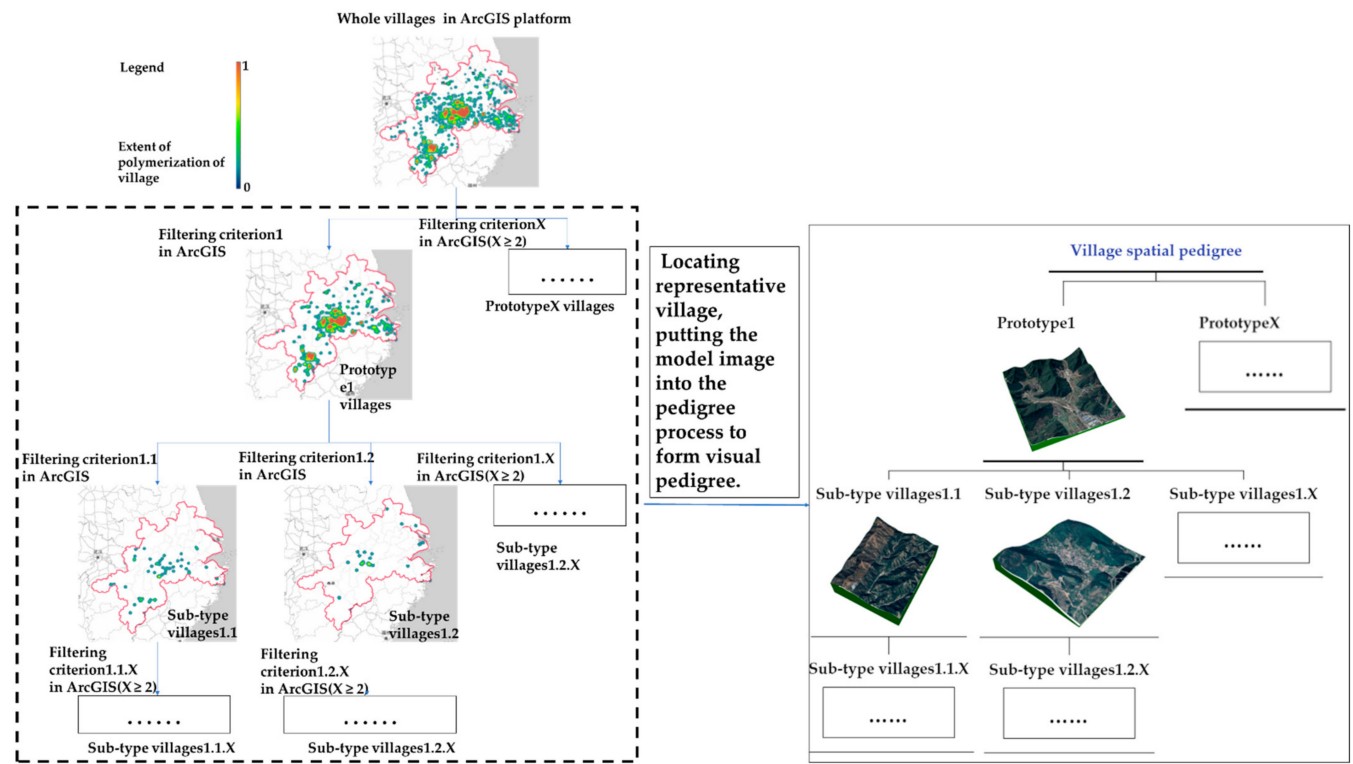

**Figure 5.** The process to filter the villages and visualize pedigree.

## 4. Results

### 4.1. Village Spatial Aggregation Patterns and Influencing Factors

#### 4.1.1. Aggregation Patterns

In order to analyze the aggregation mode of traditional villages in the Jiangnan region, this paper abstracts the villages into point elements, and conducts spatial statistical analysis and visualization in ArcGIS (Figure 6). Generally, traditional village point elements have three types of spatial distribution modes: clustering, uniform, and random, which are determined by the nearest neighbor distance and the NNI. This paper uses the average nearest neighbor distance in ArcGIS10.3 spatial statistics tools, in terms of the NNI and KDE, and the traditional villages in the Jiangnan region are characterized by local aggre-

gation and partial dispersion in space. This agglomeration feature can be summarized in the following aspects: (1) the distribution of villages has obvious circularity agglomeration characteristics; (2) there are regional differences in the distribution of villages, and the pattern of aggregation and dispersion appears in the Jiangnan region; and (3) the distribution of villages shows three distinct agglomeration cores, including two core zones in southern Anhui and one core zone in northeastern Zhejiang.

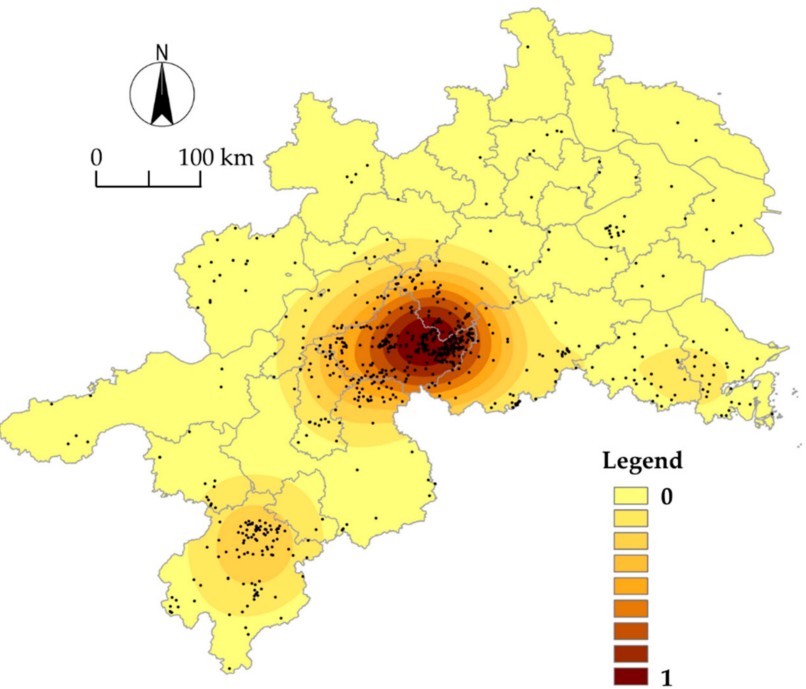

**Figure 6.** Aggregation patterns in Jiangnan region: edited by authors.

4.1.2. Influencing Factors

In this paper, the reclassification function in the spatial analysis module of ArcGIS is utilized to classify the DEM raster data according to the characteristics of Jiangnan's natural features. Then the village patch distribution layer and the geological feature layer are superimposed. The results indicate that the spatial aggregation pattern of traditional villages in the Jiangnan region is influenced by elevation, water system, and road system (Table 6). Foremost, elevation is a natural element, and it is a significant factor affecting the distribution of traditional villages in the Jiangnan region. Elevation is a limitation and a support condition for village spatial agglomeration. In general, the lower the number of villages, the higher the altitude. However, there are many villages in the higher altitude areas in the Jiangnan region, especially in the mountainous areas of southern Anhui and eastern Zhejiang [61,62]. From the perspective of Jiangnan history, this part of the villages was formed by the settlement of local ethnic minorities or the migration of historical wars to the south [63]. Therefore, the mountains have a natural sheltering effect and have a positive role in promoting the development of the countryside.

Next, studies have shown that villages related to water systems account for 73.17% of the total villages in the Jiangnan region. The water resources are natural environments formed by rivers, lakes, and oceans, which around the traditional village provide the necessary water condition. Meanwhile, agriculture was the essential life support for traditional villages in Jiangnan before the 20th century, and is dependent on the water system [64]. Therefore, in the traditional village space in the Jiangnan region, corresponding elements such as bridges, ancient wells, and wharfs have been formed because of the water system, creating a unique water town culture.

**Table 6.** Influencing factors of village spatial aggregation pattern: edited by authors.

| Elevation | Water System | Road System |
| --- | --- | --- |
|  |  |  |

In addition, traditional villages in the Jiangnan region had frequent business contacts because of the road crossing between the north and the south since ancient times [65]. Studies have shown that the villages were relying on the advantages of traffic location, causing the villages to follow or expand along the direction of the road. From the perspective of village aggregation, the road is an active area in the process of village aggregation, expansion, and development in the Jiangnan region. The greater the number of villages, the higher the degree of agglomeration, and the closer to the road. On the contrary, the fewer the number of settlements, the lower the density of settlements, and the farther from the road. It indicates that the spatial agglomeration of traditional villages in the Jiangnan region is a linear pattern distributed along the road, showing the directionality of the road.

*4.2. Traditional Village Spatial Environment Pedigree*

The Jiangnan region is in the East China Plain. Influenced by the interlacing of the Yangtze River system and the hilly terrain, the region presents a complex and flexible landform, including plains, mountains, and basins [66]. This paper uses ArcScene, ArcGIS, ArchiCAD, etc., to model and summarize the following principles: the spatial environment form of traditional villages in the Jiangnan region is dominated by mountains and rivers, and natural ecology is the principle forming three spatial environmental pedigrees of water, mountains, and mountains with rivers (Figure 7). The pedigree respectively includes the prototypes and sub-types under the influence of different terrain conditions.

4.2.1. Water System Pedigree

There are four sub-types in the water pedigree of villages: water cross the village, water surround the village, water face the village, water scatter in the village. In the Jiangnan region, 10.85% of villages are built along the water, which are scattered in the plains of the whole region, and are represented by southern Jiangsu and northeastern Jiangxi to form the gathering center. This area is an alluvial plain formed by the lower reaches of the Yangtze River, with many lakes and rivers, featuring a developed water system and a humid climate [67]. In this pedigree, village space is dominated by the water system, and the road relies on the river network, which created the water environment and can adjust to the changeable weather. The villagers formed a powerful irrigation and transportation system by utilizing the abundant water resources, arranged the water network and roads in accordance with the water system, and formed a wharf with frequent foreign trade exchanges.

4.2.2. Mountain System Pedigree

There are four sub-types in the mountain pedigree of villages: mountain surround the village, the village sprawl mountainside, the village entrenched on the ridge, the village scattered among the mountains. There are 26.92% of the villages built on the mountains

in the Jiangnan region, which built the vernacular architecture with the wood and soil from mountain resources [68]. They are distributed in the hilly and mountainous areas of southern Anhui, and a few are scattered in Jiangxi and northeastern Zhejiang. The reason is that the natural sheltering function of the mountain can ensure sustainable development. Mountain villages such as Mu Ligong Village and Xumin Village have historically avoided disasters and people escaped to the Jiangnan region, where they built villages to breed clans. The indigenous people in this area, including the She nationality, have grown up in the mountains and forests for a long time to preserve their primitive life, and are uneasily affected by urbanization. The villages are built on the mountain, which is complex, changeable terrain and diverse. Consequently, the village space required higher construction skills, and the natural protection enables Jiangnan cultural to be inherited and continued [69]. Additionally, the traditional villages built on the mountain reflect the traditional ideology of respecting nature because of the complexity, difference, and closeness of mountain conditions. In the process of long-term adaptation and coordination with nature, a variable pedigree structure has evolved.

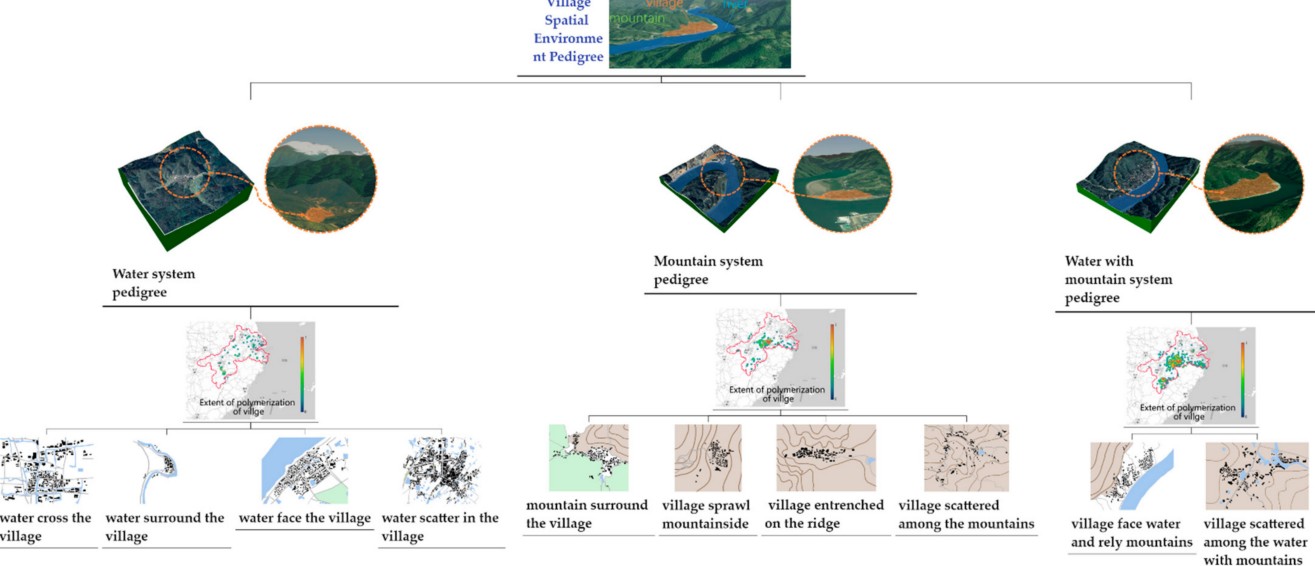

**Figure 7.** Traditional village spatial environment pedigree structure: edited by authors.

### 4.2.3. Water with Mountain System Pedigree

There are two sub-types in the water with mountain pedigree of villages: village face water and rely mountains, village scattered among the water with mountains. There are 62.32% of villages in the water with mountain type, which is the highest proportion of the environment forms in the Jiangnan region. They are distributed in areas with both mountains and water bodies, and scattered in various parts of the Jiangnan region, indicating that this is a tendency of group consciousness for constructing traditional villages which is beneficial to develop the local life. The water system is not only a river course, a traffic system, and a business system, but also has the functions of drainage, fire prevention, and climate regulation. The mountain is a windbreak, hedging, and a defense system, creating a unique regional human settlement model. The combination of landscapes has formed the most important environmental pedigree of Jiangnan villages, and integrated the advantages of the natural resources to form the village Fengshui culture [70].

### 4.3. Traditional Village Spatial Organization Pedigree

The spatial organization refers to the spatial distribution and configuration method formed under the influence of the natural environment as the dominant factor (Figure 8). The form intuitively shows the result of a dynamic extension of the village space to adapt

to the environment [71]. The types and changes of the spatial organization reflect the long-term aggregation of the clans living in the culture. Based on village survey and draw, and on-site investigation, the paper uses ArcScene, ArcGIS, ArchiCAD, etc., to model and analyze, and synthesize the ways of villagers' group living. According to the field survey data, we use ArchiCAD to mark and identify the internal public space elements of the village, the road water system, the village axis, and the architecture group. Then we classify and quantify the qualitative data of the same kind of village space to confirm the criteria used to recognize the village spatial organization pedigree (Table 4). The pedigree of the spatial organization of traditional villages in the Jiangnan region is divided into three aspects: centripetal aggregation pedigree, net radial pedigree, and random scatter pedigree.

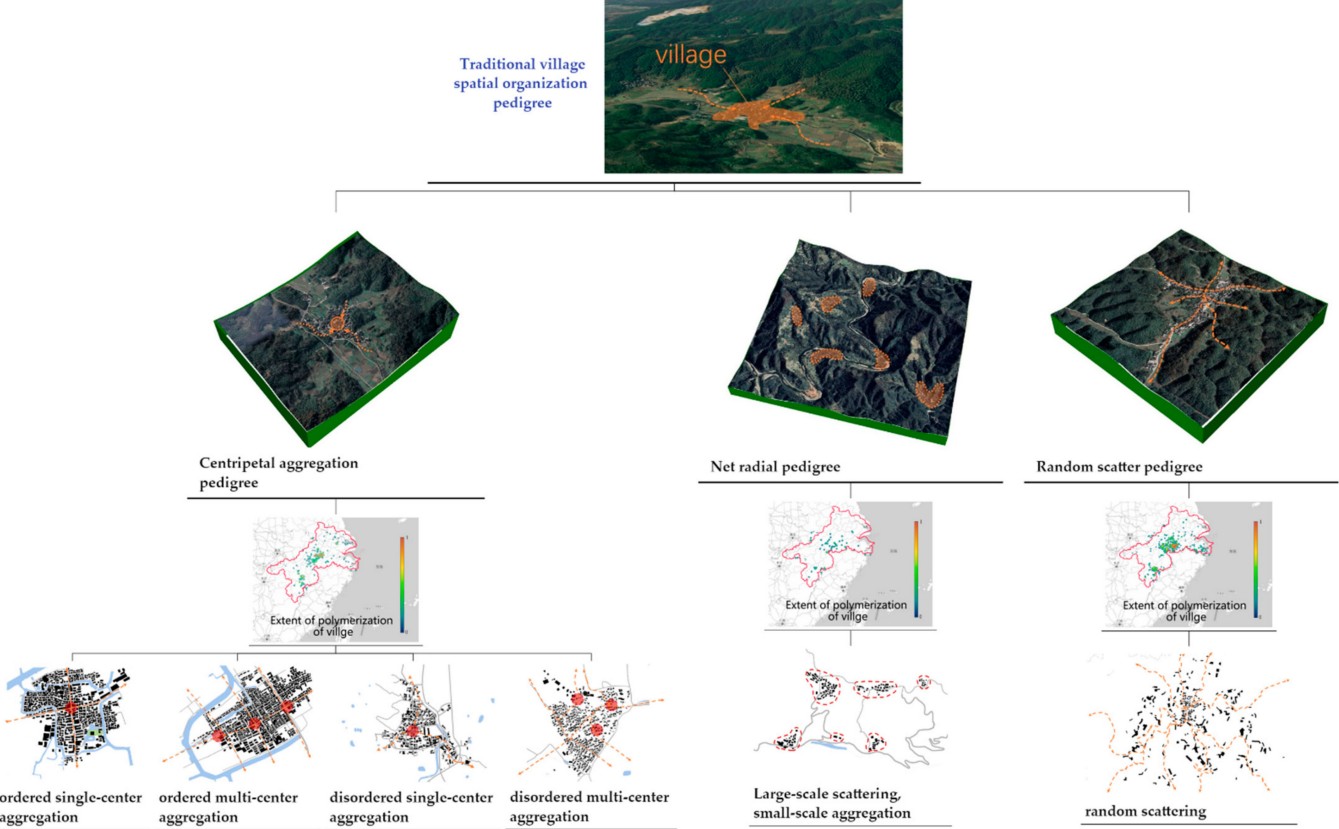

**Figure 8.** Traditional village spatial organization pedigree structure: edited by authors.

4.3.1. Centripetal Aggregation Pedigree

This type of pedigree is divided into single-center aggregation and multi-center aggregation modes. This spatial pattern forms one or more village cores with public activity spaces such as squares, ancestral halls, temples, theatre stages, and natural landscapes, etc. Generally, the center aggregation has a special spiritual meaning and group cohesion within the village [72]. The centripetal aggregation is created in the villages with scale and dominant power [73]. The core of the spatial pattern of the village is obviously directed. The village groups and the system of streets and alleys are developed around the core. The overall form is closely arranged around the core and the levels are clear. Centripetal cohesion of the group is the embodiment of the ethnic centripetal force, which present as public architectures in villages The core of the village is clear, but the spatial form is irregular in a special area. It is present in flat or steep terrain, with different scales and free aggregation forms, indicating that local aggregation forms adapt to external natural conditions through technical construction. The centripetal aggregation pedigree of villages is the choice of the family and clan group living mode under traditional farming production, and it is affected by the effects of social factors such as blood, geography, and industry [74].

### 4.3.2. Net Radial Pedigree

Traditional villages of this pedigree type radiate in disorder along roads, water systems, or mountains. The space is restricted by the natural topography, which leads to the weakening of the aggregation core of the village, showing the morphological characteristics of outward spreading and radiation in the form, similar to a spread out net skillfully covering the natural landscape. The roads and water systems in such villages are planning axes of the villages, and the spatial pattern of roads is distinct. The net radiating form increases the outward connectivity of the village space. Due to the outward growth of the reticulated and radial villages, the scale of the village is proportional to the population. The convenience of roads improves the village's external communication, thereby absorbing new forces and improving the quality of the village, which are very conducive to foreign trade and cultural integration [75].

### 4.3.3. Random Scatter Pedigree

This pedigree type of villages is distributed in areas with severe terrain or climatic conditions in the Jiangnan region. Due to limitations of topographic and geomorphological conditions, villages random scatter in the flat landform. Under naturally severe environmental conditions, the villages are connected by winding roads, and small-scale gatherings are carried out in suitable topographical conditions, presenting a pattern of scattered layout of the village core. The characteristics of the village indicate a large dispersion and small aggregation, and the land between the architecture is narrow [76]. In addition, the road system of these villages is scattered and loose, and the boundaries of the villages are large and blurred. The internal connection of the village is weak, the houses are relatively independent, each area is independent, and the public activity space is scattered.

### 4.4. Traditional Village Spatial Morphological Pedigree

Due to the spontaneity of construction, the spatial form of a traditional village in the Jiangnan region ranges from singularity and functionality to diversity, contingency, and flexibility. It makes the village space break away from the solidified spatial pattern and has local cultural connotation and vernacular characteristics [77]. Based on village survey and draw and 3D modeling by ArcScene, ArcGIS, ArchiCAD, etc., this paper summarized village space through typology. According to the field survey data, we used ArchiCAD to mark and identify the village outline boundary style, the road water system, the village axis, and the number of village units. Then we classified and quantified the qualitative data of the same kind in the village space to confirm the criterion used to recognize the village spatial organization pedigree (Table 5). The pedigree of the spatial morphology of traditional villages in the Jiangnan region is divided into three aspects: regular form, irregular form, and artistic form (Figure 9).

### 4.4.1. Regular Form

Regular form, also known as geometric form, reflects the view of formal rules and a sense of strict order. The axis and center of the village space are the visual core, which strengthens the inner cultural force of the rural community. The type of pedigree follows traditional thought about etiquette and was formed under the influence of the northern Han culture. The rules and forms echo the traditional social hierarchy and moral and ethical consciousness, which are the rational aesthetic expression of the Jiangnan village space [78]. Village craftsmen create distinguished spatial forms through symmetry, translation, rotation, reflection, etc., and use axis and ritual planning to strengthen the majesty of local clan autonomy, with highly generalized symbolic and artistic effects. Nevertheless, in the Jiangnan region, due to the changeable water systems and complex topographical conditions, the regular form of traditional villages requires the flat level of terrain. The proportion of villages with regular forms is low, and the geometric forms are formed by the joint action of terrain and ideology.

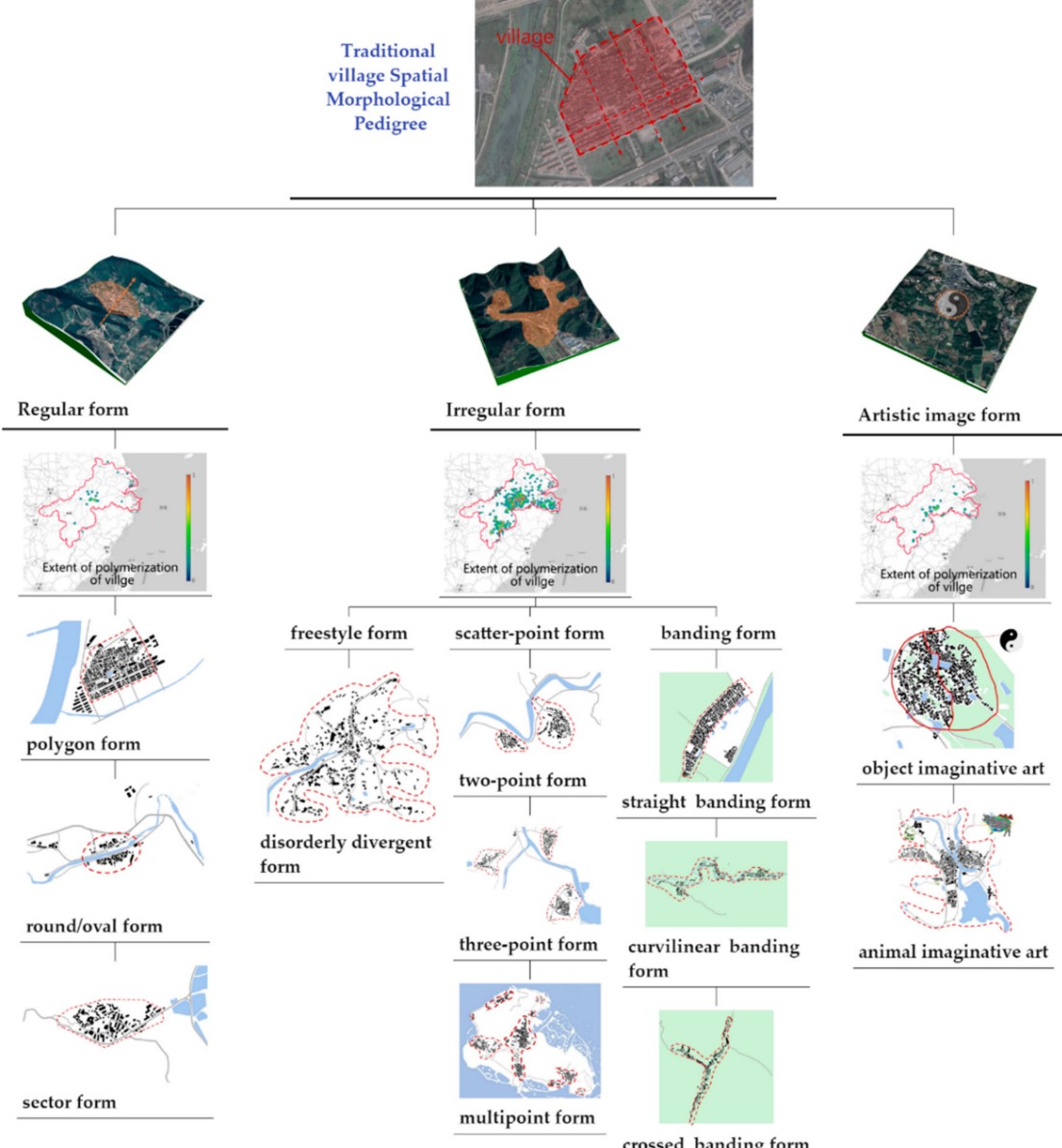

**Figure 9.** Traditional village spatial morphological pedigree structure: edited by authors.

### 4.4.2. Irregular Form

Changing water systems and mountains in the Jiangnan region have created unpredictable natural conditions [79], which led to the largest proportion of village forms being irregular:

(1) Freestyle form: The spatial form of the village expands along the surrounding mountains, water systems, and road network space with the expansion of the population, forming a random free form. The extension form of space is flexible and full of rhythmic changes, while the spatial morphology of the village is based on natural landforms and is assembled and arranged flexibly under the invisible blood ties.

(2) Scatter-point form: This form is divided into three sub-types, including two-point form, three-point form, and multipoint form. The villages are dispersed in different landforms in the Jiangnan region. In terms of the severe terrain within the village area, village space is distributed unconsciously. Therefore, the village space is scattered in the flat area for construction by one or several units, forming the scattered form.

(3)   Banding form: These villages are morphologically banded, forming two subtypes, including straight banding form and curvilinear banding form. The straight banding form extends along both sides of a main street or river, and the street layout is connected by one or two main street axes running through the space. The village plan is arranged in a straight line along the winding streets, highlighting the directness and penetration of the banding shape. The curvilinear banding forms a spatial banding-shape for the village layout to spread along the parallel contour lines of the mountains. The form of village space presents a changeable line, and the depth is arbitrary. It is built by craftsmen according to the need of the villagers, forming a multi-level linear extension.

### 4.4.3. Artistic Form

The artistic form is a special morphology created by extracting the essential features of the village space and refining, enriching, and connecting the corresponding form types with cultural and social implications [80]. The idea of pictograms took shape in the early creation of oracle bone inscriptions, and by absorbing experience in nature to obtain references, and then moved towards artistic space design [81]. The artistic form is divided into object imaginative art and animal imaginative art. Artistic form observes and learns objects from nature, and creates and imitates them through production, showing the characteristics of the form of things. Meanwhile, it presents very specific symbolic graphics or cultural semantics through imaginative refinement and interpretation. Artistic form does not mean complete copying, it is the depiction and rendering of natural scenery with emotional imagination in traditional social life, and it is the condensed appearance of specific objects.

### 4.5. Traditional Village Architectural Form Pedigree

The study collects and organizes the information of architectural forms in the village space in 728 villages of the Jiangnan region, inputs the data in ArcGIS, and constructs the models. Then, it assigns attributes to each village space in the models, compares and analyzes the spatial areas in ArcGIS, and sorts out the forms of similarities and differences. According to the time context and type hierarchy, the architectural form pedigree is sorted out. From the perspective of vernacular architecture, the pedigree was impacted by the regional diversity, as summarized in five aspects: Shanghai residence, Southern Jiangsu residence, Southern Anhui residence, Northeast Jiangxi residence, and Northeast Zhejiang residence (Figure 10).

### 4.5.1. Architectural Form

Shanghai residence has five sub-types which had inherited the courtyard-style building type influenced by northern construction technology [82]. Due to architectural forms in the Shanghai area being impacted by Western culture since modern times, they have evolved into a unique sub-type combining Chinese and Western cultures. There are six sub-types of residence in southern Jiangsu, showing the integration with the water environment, and diverting water into the indoor space. Simultaneously, this area might be used for a gathering with a history scholar or celebrities, so that Su-style residences are flexible and integrate traditional garden artistic details [83]. There are four sub-types of residence in southern Jiangsu. Residential houses in southern Anhui are affected by the water system environment and the mountainous terrain, and the sub-types include courtyard-style and its variants. Its construction technology focuses on the use of regional materials to form the typical Huizhou residence [84]. There are five types of residence form in northeastern Jiangxi, which reflect the compatibility under the influence of distinguishing architectural styles and cultures. Architectural types include Hui-style buildings in southern Anhui and waterfront residence in southern Jiangsu, showing similarities in construction techniques [12]. There are 10 sub-types of residence in northeastern Zhejiang, which present unique spatial forms and construction details. The architectural space is combined and changed because of the courtyard landscape, which reflects the requirements

of the expansion of the building scale and the change of the shape under the need of human settlements [68].

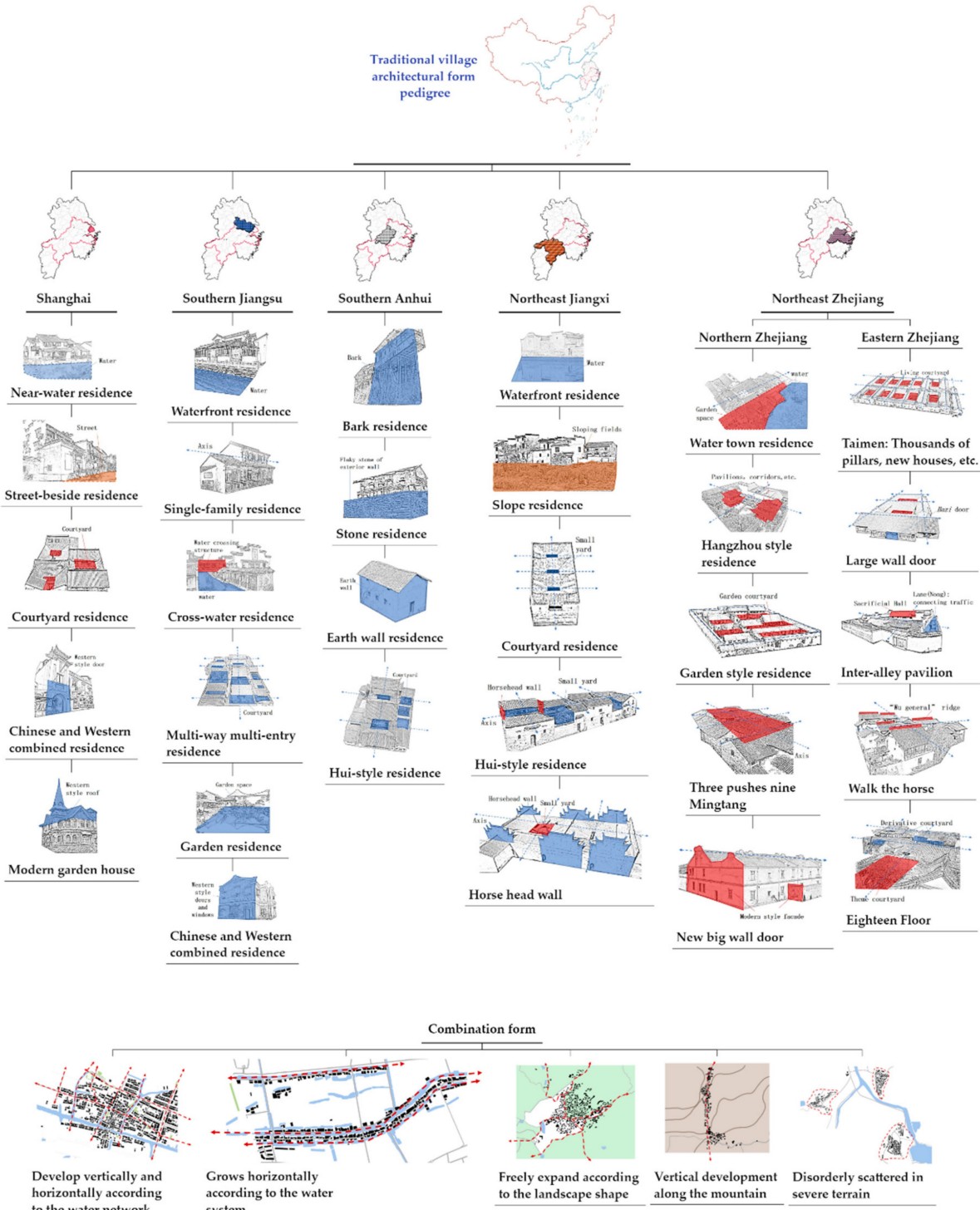

**Figure 10.** Traditional village architectural form pedigree structure: edited by authors.

From the perspective of time and space evolution, architecture types can be mainly divided into three levels: (1) combining with the natural environment, using the natural materials such as stone, wood, bark, and other raw materials for construction; (2) a courtyard model dominated by the idea of harmony culture [85]; and (3) blending with foreign and spontaneous cultures, new architectural models have evolved, including a

garden culture, Western culture, and local traditional culture [86]. From the perspective of space type, regional differences create distinctive architectural forms, which are as follows: different name forms for the same type of building are influenced by regional culture and language, and the names and expressions are slightly different, but they are essentially one type of architectural style. For example, water-related residences are called near-water residences in Shanghai, waterfront residences in southern Jiangsu, and water town residences in northern Zhejiang. The same architectural forms in different regions, such as Huizhou residence, have architectural forms in southern Anhui and northeastern Jiangxi. Historically, the architectural styles formed by the mutual trade between Huizhou merchants and Gan merchants exchanged with each other [87]. This is reflected in the details and decoration of the building and the aesthetic preference in the sculpture art, which is embodied in the combination of the Chuandou (post beam and tie framework) and the Tailiang (post beam and strut framework), as well as the building materials made of wood and masonry [88,89]. The similarity of architectural forms is reflected by the influence of common vernacular Jiangnan culture, such as garden culture, which is expressed in the structure of the residence, courtyard landscape, and architectural details by drawing on the elements of landscape and water [90].

### 4.5.2. Architectural Group Form

There are five forms of architectural group in Jiangnan traditional villages, including: develop vertically and horizontally according to the water network and roads; grows horizontally according to the water system; freely expand according to the landscape shape; vertical development along the mountain; and disorderly scattered in severe terrain. Its advantages include the following aspects: (1) The combination of architectural groups conservates land and adapts to the terrain [91], it satisfies living and production requirements of the clan group under limited conditions, and it facilitates the expansion of the scale formed by the development of the population and economy. (2) Reverence for ecological concept minimizes the damage to nature by means of science and technology. It constitutes a characteristic model integrating production, residence, and tourism with local culture. This model is in an enlightening effect on today's cultural tourism development [92]. (3) Combining the cultural and topographical conditions of the north and the south, and driven by the internal drive of population expansion, a continuous and repetitive architectural group organizational model has been formed, which is convenient for local groups to live and manage uniformly. The wholeness of the village space is realized by the dynamic coordination and unity of individual characteristics, but in suitable situations it shows the characteristics of "harmony but different" due to the integration of peoples [93].

### 5. Discussion

*5.1. On-Site Mapping and 3D Modeling Application of Traditional Village Space*

In terms of research area, traditional villages in the Jiangnan region are in the rural areas of China's economic development belt since ancient times, and their rich environmental patterns have a significant impact on the evolution of urban-rural space [94]. This area is a typical region of traditional Chinese culture. The time span is from the Tang and Song dynasties to the present, and the data of 728 villages have formed a quantitative research sample basis. Due to the complexity of its natural environment and the long historical span, the images and models of the village space cannot be directly obtained from the existing data, and they needed to be obtained through field investigation. Although technologies such as machine learning or remote sensing tools can identify villages from satellite images, they cannot accurately identify spatial layers such as boundaries, architectures, and landscapes in the village space. In this research, through the method of on-site surveying and mapping, we made up for the lack of specific spatial information on the traditional villages in the Jiangnan region in areas which are remote or isolated. Through the layered transformation of BIM into ArcGIS software, the spatial information obtained by on-site investigation can accurately generate a 3D village model, and its scale accuracy can reach

1:1, which is helpful for the classification of village spatial layers and accurate identification of space. This method can be used for accurate spatial modeling of such isolated area space. Village spatial information presents the diverse features of human environment space, which is a spatial reference for inheriting local historical heritage, and expands the limit cognition of urban historical space. Meanwhile, it is an inspiration to explore the urban regeneration mode in combination with the local environment and culture information such as Fengshui principle, local worship space, art, religion, etc. It is necessary to supplement traditional village space information exploring the unique characters in urban-village conservation.

Other regions in China, such as Hunan and Guizhou, have also begun to select typical cultural regions for pedigree exploration in recent years, and we can see the typical selection of study regions and the trend of pedigree understanding [9,31,36]. Under the changing environment concept and the cultural background of mutual influence, the traditional village space in the Jiangnan region reflects the profit-seeking and inheritance of the collective construction mode [95]. Therefore, the development of village space represented by region from prototype to sub-types is the result of historical development, and it leads to the independent evolution process of different types in specific regions.

*5.2. Classification of Jiangnan Traditional Village Space in Pedigree*

The surveying and mapping data of the traditional villages in the south of the Yangtze River can be directly classified and labeled by 3D modeling to identify the types of spaces such as streets, water systems, and buildings. However, due to differences in natural environment, man-made construction, and historical context, the study found that the 728 samples of traditional villages have different appearances. Although the expressions of the village space are different, they also have spatial similarities. For example, in the four villages in Table 7, their orientations make the spaces appear different due to the terrain, yet all of them have a main street, a mountain element, and the architectures are organized in a strip along the street, shaped all similarly as a banding form. Based on the spatial features of similarity, we construct the pedigree by utilizing ArcGIS and BIM, doing that to classify criteria by spatial overlay analysis. In this research we focus on the ArcGIS character which can display and analyze the village spatial features at macro, meso, and micro levels. Through analyzing the distribution and geological features of villages, we obtain three main influencing factors at the macro level. Then, according to the three main influencing factors, we divided the spatial data according to elevation, water system, and road system of the survey from meso level and micro level in ArcGIS by SOA. In terms of the meso level and micro level clues, we obtain four criterion frameworks for recognizing the types of village spatial pedigree. Then we use this criterion framework input to ArcGIS, and utilize the filter command to classify the prototypes and sub-types with criteria at different levels. The link between prototypes and sub-types constructs the pedigree which reveals the cross-correlation between multiple villages.

In terms of research subjects, the evolution and distribution of types in the spatial pedigree of traditional villages in the Jiangnan region are the result of the combined action of multiple factors. The environment is an external factor that affects the aggregation and change of types. Culture, residents' activities, and local architectural skills are the intangible factors which affect the classification of typical pedigrees. The research results indicate that the expression form of Jiangnan traditional village space has the following internal logical relationship (Figure 11): (1) The village environment reflects the natural ecological relationship, and creates a suitable human settlement environment by consciously changing the spatial form or the degree of aggregation in the changing conditions of impermanent external forces. (2) The central axis symmetry is applied to the village spatial layout structure and street planning system, but subject to a changeable water system and topographical environment, it forms a flexible but distinct spatial form. (3) The village spatial technique integrates the northern construction techniques, and solves practical problems such as functional distinction and land saving, and creates a new form suitable

for the local area [96]. (4) Through commerce, education and other activities, the space art form has been transformed from aristocratic to folk, such as ancient pagodas, academies, feng shui and other elements which are comprehensively used in the village space. It has formed the characteristics that the village space presents with similarities in a large range and with cultural commonalities, and individual villages in various cultural belts present the characteristics of differences and independence.

**Table 7.** Similarities and differences in example of banding form villages: edited by authors.

| Village Name | Qi Ling Village | Yang Wan Village | Shi Chuan Village | Su Village |
|---|---|---|---|---|
| Models | | | | |

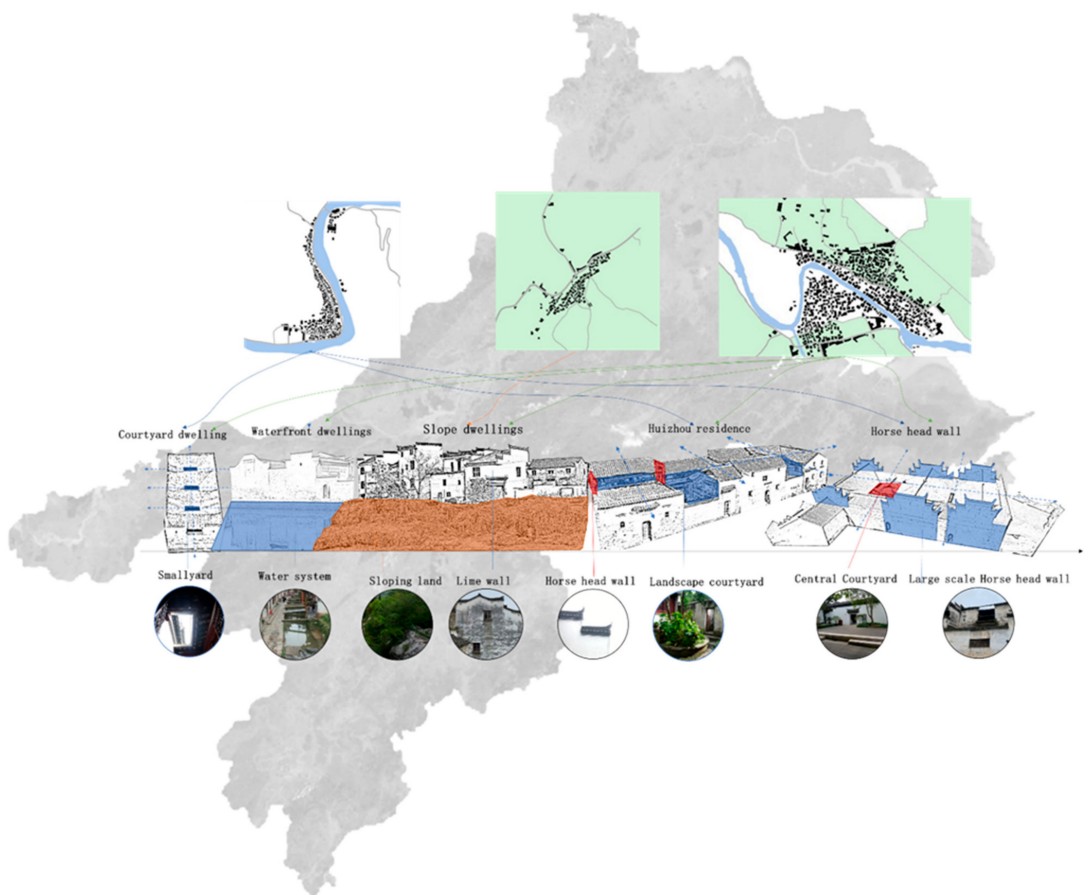

**Figure 11.** Jiangnan traditional village internal logical relationship: edited by authors.

### 5.3. The Application of Traditional Village Pedigree for Urban Regeneration

Due to ecological natural landscape and diversified traditional historical architecture, the traditional villages in the Jiangnan region can bring multi-level positive effects to the urban areas in the principle of protection and reuse. These villages are developing in coordination with the urban areas in different functions, and play a role in activating the vitality of the urban areas from the perspective of conservation, which including the use of village tourism, traditional industry empowerment, cultural inheritance, and other models

to activate urban regeneration (Table 8). Therefore, it is necessary to provide reasonable protection and reuse methods for different types of villages. In this study, the focus is to provide a reference for the transformation and reuse of villages from the perspective of urban regeneration by sorting out the types in the pedigree. Its positive effects can be summarized as the following: (1) Through on-site research, the spatial information and model of traditional villages in remote areas was obtained and constructed, which made up for the spatial simplification conservation method caused by insufficient information. (2) Combining the terrain analysis capabilities of ArcGIS and the accuracy of BIM modeling to complete 3D modeling, it can be used for the stratification of village spatial information and the classification of types, which are convenient for visualizing spatial information of historical conservation in urban regeneration. (3) Pedigree establishes cross-correlation of villages in multi-source spatial features. Village information is no longer isolated and singular information. The pedigrees and database established in the research facilitates the digital management of village conservation in urban regeneration. (4) Village pedigree construction helps to classify traditional villages, which can be helpful for choosing a method to reuse. For instance, same type or area villages can focus on similar reuse methods. Moreover, differentiated strategies can be selected for different village types based on classified, rather than a uniform or single, conservation standards to inspire the diversity of urban-village space.

The method to construct the pedigrees in the Jiangnan region requires more complete village information and visualization. For villages in isolated areas without information, we have established a set of spatial modeling methods: First, survey and map the village spatial data on-site, use ArchiCAD to draw the model, and conduct on-site inspection and review. After that, use ArchiCAD to label village streets, water systems, architecture, etc. Then, transform the model format through the global mapper and input it into ArcScene, and superimpose it with the village spatial information to generate a 3D model. Pedigree research intuitively indicates the composition of village space construction technology and the influence of time development, and is an exploration method that integrates typical regional cultural genes in village space [97]. The study of classification and pedigree construction is the basis for analyzing the spatial relationship between villages and urban space. Expansion and classification of basic data help to understand the formation, stability, adjustment and change of types, and are an essential perspective for the discussion of contemporary urban-rural co-progress. In addition, the application of ArcGIS and BIM transforms traditional information or digital research into visual model analysis, which has positive significance for the intuitive understanding of the villages as a space [10]. At the present stage, the construction of the model requires a lot of on-site research, and it points to the lack of basic data on resources in contemporary traditional villages. Therefore, it is necessary to supplement it continuously with the development of the urban space. The method of construction pedigree contributes to form a characteristic regional research database.

**Table 8.** Urban regeneration types of traditional villages in Jiangnan region: edited by authors, information is obtained from field research, photograph taken by authors.

| Types | Village Representative | Positive Effect on the Urban |
|---|---|---|
| Diversified industry | 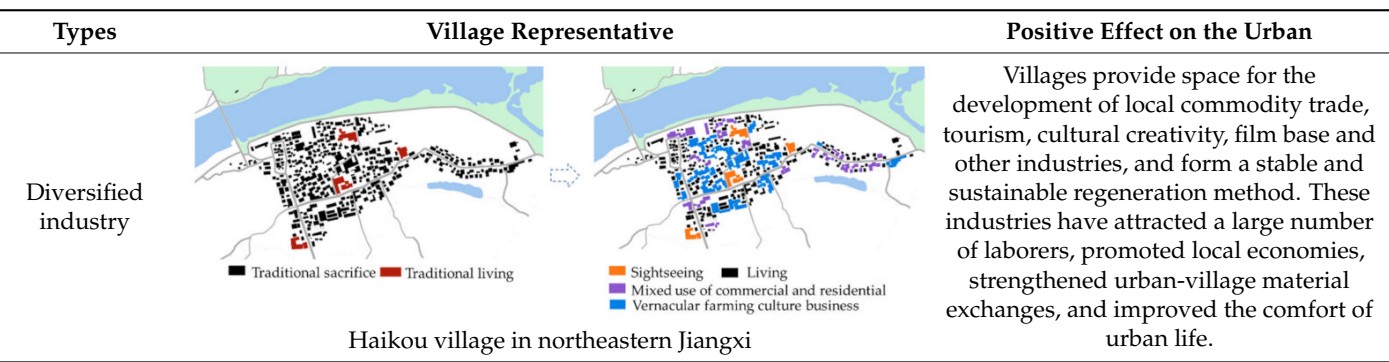 Haikou village in northeastern Jiangxi | Villages provide space for the development of local commodity trade, tourism, cultural creativity, film base and other industries, and form a stable and sustainable regeneration method. These industries have attracted a large number of laborers, promoted local economies, strengthened urban-village material exchanges, and improved the comfort of urban life. |

**Table 8.** *Cont.*

| Types | Village Representative | Positive Effect on the Urban |
|---|---|---|
| Cultural heritage | 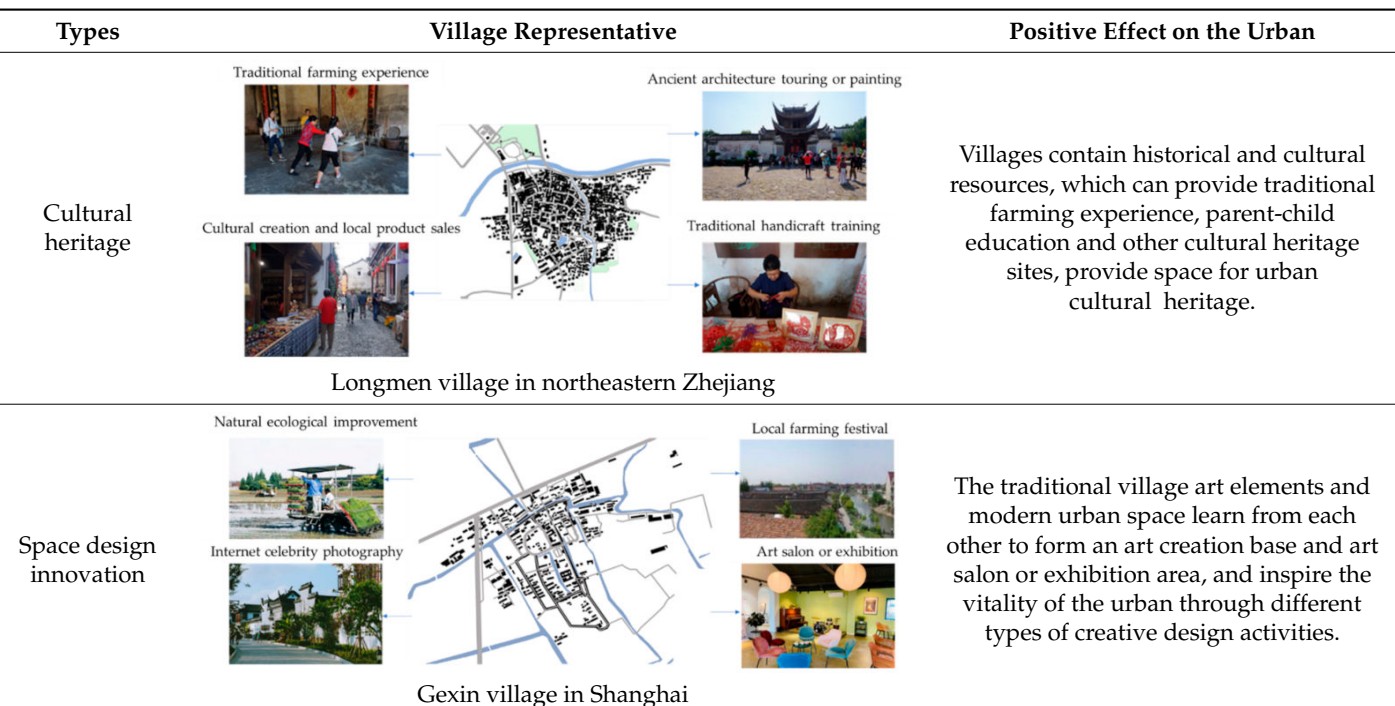 Longmen village in northeastern Zhejiang | Villages contain historical and cultural resources, which can provide traditional farming experience, parent-child education and other cultural heritage sites, provide space for urban cultural heritage. |
| Space design innovation | Gexin village in Shanghai | The traditional village art elements and modern urban space learn from each other to form an art creation base and art salon or exhibition area, and inspire the vitality of the urban through different types of creative design activities. |

## 6. Conclusions and Ongoing Research

The spatial aggregation pattern of traditional villages in the Jiangnan region is affected by elevation, water system, and roads system, presenting three agglomeration cores. The spatial pedigree of traditional villages in the Jiangnan region is divided into four contexts from the perspective of spatial types. The three sub-types of traditional village spatial environment pedigree present the mechanism by which village space adapts to the changes of the natural environment. The three sub-types of traditional village spatial organization pedigree indicate the effect of settlement living consciousness and spatial layout. The three sub-types of traditional village spatial morphological pedigree are a manifestation of the interaction between the construction logic of space and the environment, society, and local imagination. The five sub-types of traditional village architectural form pedigree show the local contextual structure of region, culture, and construction skill. These four pedigrees illustrate prototypes and sub-types of spatial forms under the various factors, which are analyzed and realized through spatial overlay and visualization models. The diversity of the spatial pedigrees of traditional villages in the Jiangnan region originates from the diversity of rural life, including both the tangible and intangible. The superposition of these factors shows the logic of pedigree by visualization, which can be considered as cultural commonality and regional characteristics. From the perspective of the method of historical protection in urban renewal, village information in remote areas needs to be acquired by on-site surveying and mapping. Attention should be paid to the study of this type of space by this method, which can supplement the missing qualitative information of the space type, which is also the basis of the three-dimensional modeling of the village. Meanwhile, although the combination of ArcGIS and BIM tools can accurately generate models to identify village space, the sorting of village space types requires spatial attributes in hierarchical architectural models, and this part needs to rely on the criterion of qualitative data. Additionally, the essence, advantages, and disadvantages of traditional regional design are explored from the spatial pedigree of traditional villages in the Jiangnan region, which is conducive to the extraction of spatial elements. It could be applied to the process for inheritance and reuse of village characteristics, and we also can use spatial elements as a breaking point to explore the possibility of future urban regeneration.

However, as mentioned above, the data of traditional villages requires much on-site work, so there is still insufficient research on the aspects of occupant behavior, village microclimate, building comfort, social and economic conditions, etc. In the future, to understand the correlation and development of pedigrees thoroughly, ongoing research activities should include: (1) Based on the big data model, specifically the analysis of the spatial morphology of the village, including residential structures, artistic details, technical methods, etc., extracting complete protection and urban regeneration methods from it. (2) Continue to use instruments to supplement the data of sound, light, and heat of villages through field investigations, and improve and extend the content of pedigrees. By integrating or simulating data resources, such as current life behavior patterns, we can build a foundation for protection, inheritance, innovation, and transformation of the village environment. Moreover, it can be used for future comparisons to other regions to build distinctive regional databases. (3) Based on spatial pedigrees to determine which spatial characteristics are necessary and unmodifiable, and which spatial characteristics can be weakened and unified, future machine learning can be used to improve the accuracy and efficiency of spatial recognition and labeling.

**Author Contributions:** Conceptualization, X.L. and Y.W.; methodology, X.L.; software, X.L.; validation, X.L., Y.L. and C.L.; formal analysis, X.L.; investigation, X.L., Y.L. and C.L.; resources, Y.W.; data curation, X.L.; writing—original draft preparation, X.L.; writing—review and editing, X.L. and Y.L.; visualization, X.L. and C.L.; supervision, Y.W.; project administration, X.L.; funding acquisition, X.L. and Y.W. All authors have read and agreed to the published version of the manuscript.

**Funding:** This research was funded by "Research on the protection and development of traditional village buildings in Jiangsu from the perspective of rural revitalization, grant number 20YSA001". It was also funded by "Research on the spatial protection design of traditional villages in Shandong Peninsula from the perspective of rural revitalization, grant number 22DWYJ28". The APC was funded by the Jiangsu Provincial Office of Philosophy and Social Science Planning, and the Shandong Provincial Office of Philosophy and Social Sciences.

**Acknowledgments:** The Ministry of Housing and Urban-Rural Construction of the People's Republic of China provided the information about traditional villages. The School of Architecture, Yantai University, provided the technical analysis platform used in this research. This study is funded by the grant from the Jiangsu Provincial Office of Philosophy and Social Science Planning (grant number 20YSA001), and the Shandong Provincial Office of Philosophy and Social Sciences. Special thanks to the reviewers and editor for very useful comments and suggestions.

**Conflicts of Interest:** The authors declare no conflict of interest.

## Appendix A

**Table A1.** The Survey and Draw Image of Jiangnan Traditional Villages Used for Model Visualization in the Article: Edited by Authors.

| Lu Zhong Village | Shi She Village | Dong Ziguan Village |
|---|---|---|

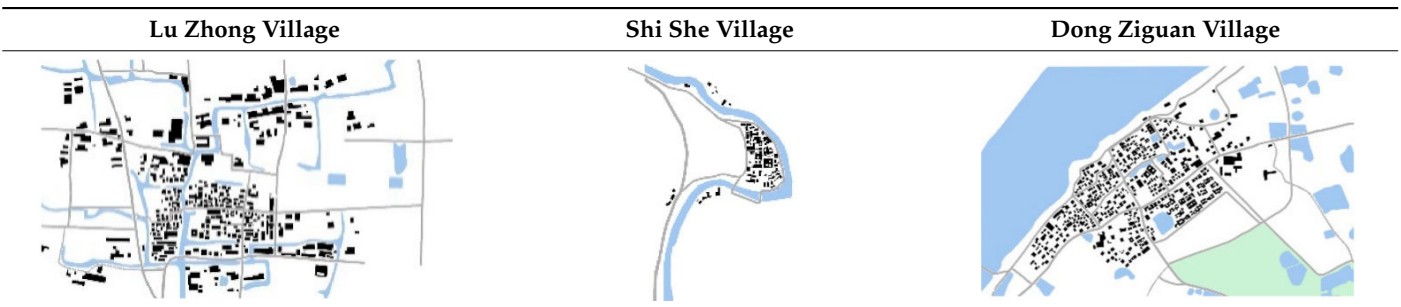

**Table A1.** *Cont.*

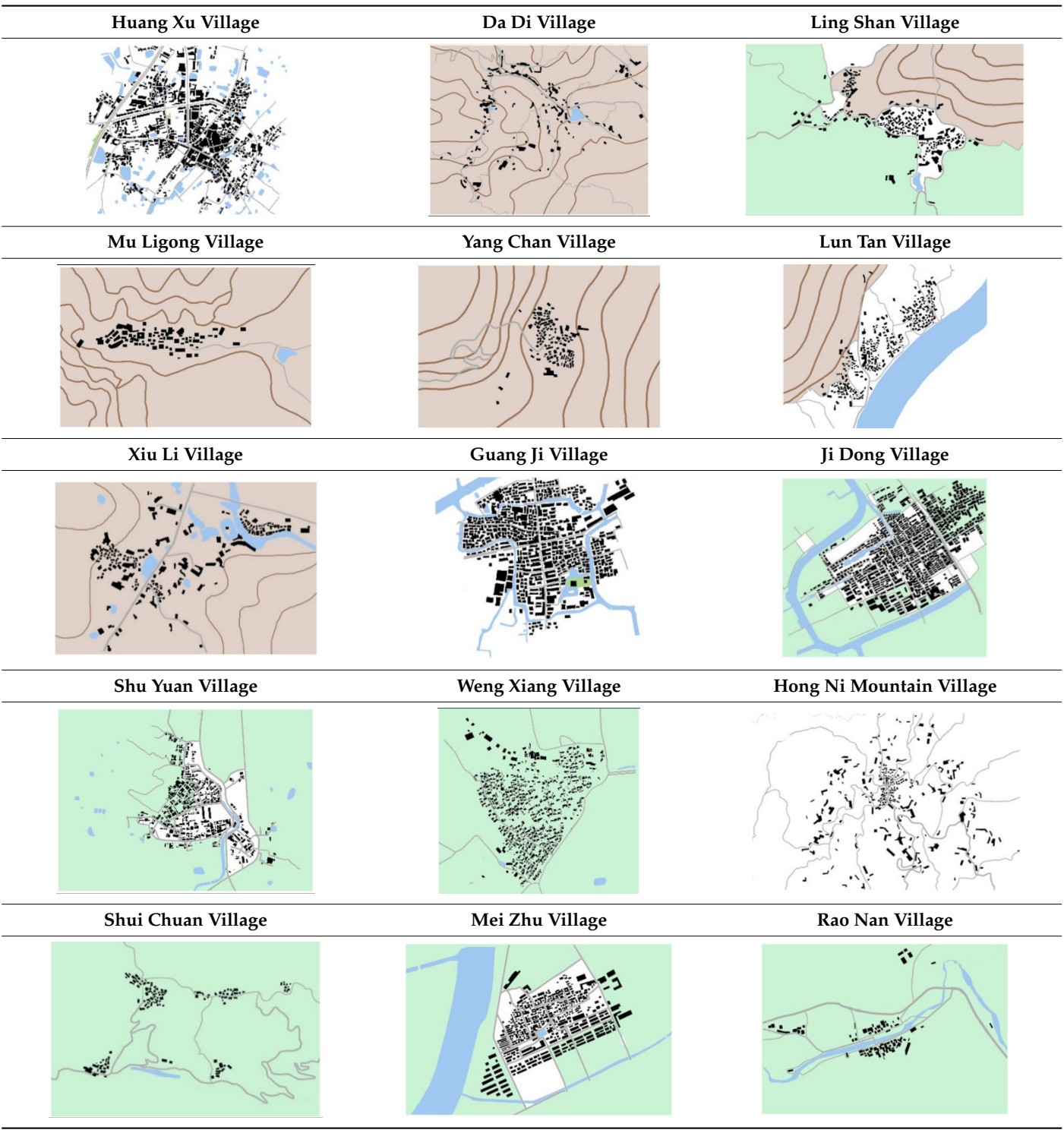

**Table A1.** *Cont.*

| Ruo Dai Village | Xiang Chang Village | Up River Village |
|---|---|---|
| San Xin Village | San Shan Village | Mei Rong Village |
| Hu Ri Village | Two Leaves Village | The Eight Diagrams Village |
| Hong Village | Qi Ling Village | Pu Xin Village |
| Yang Qiao Village | Zhen River Village | Dai Jiashan Village |

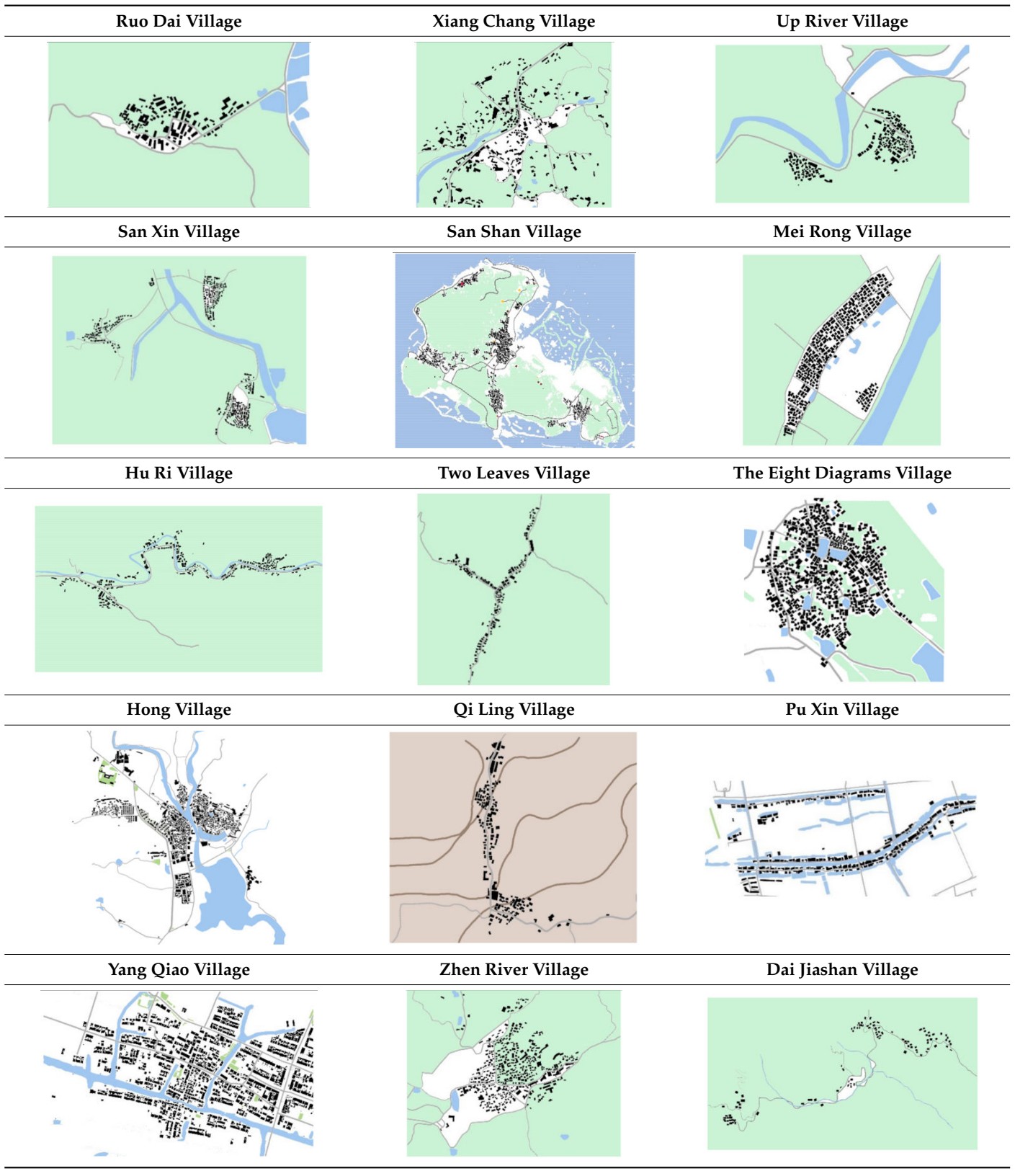

## Appendix B

**Table A2.** The Criterion Framework for Traditional Village Spatial Environment Pedigree: Edited by Authors.

| Level | Macro Level | Meso Level | | | | | | | | | | | |
|---|---|---|---|---|---|---|---|---|---|---|---|---|---|
| | | The Criterion Framework for Traditional Village Spatial Environment Pedigree | | | | | | | | | | | |
| | | Prototype | | | Mountain System Pedigree | | | | Water System Pedigree | | | | Water With Mountain System Pedigree | |
| Research Object | Influencing Factors | | Sub-Types | | Mountain Surround the Village | Village Sprawl Mountain-side | Village Entrenched on the Ridge | Village Scattered Among the Mountains | Water Cross the Village | Water Surround the Village | Water Face the Village | Water Scatter in the Village | Village Face Water and Rely Mountains | Village Scattered Among the Water with Mountains |
| Criterion 1 | Elevation | Terrain elements | relationship with mountains | mountain surround | ✔ | ✗ | ✗ | ✗ | ✗ | ✗ | ✗ | ✗ | ✔ | ✔ |
| | | | | village sprawl in | ✗ | ✔ | ✗ | ✗ | ✗ | ✗ | ✗ | ✗ | ✗ | ✗ |
| | | | | village with ridge | ✗ | ✗ | ✔ | ✗ | ✗ | ✗ | ✗ | ✗ | ✗ | ✗ |
| | | | | village scattered in | ✗ | ✗ | ✗ | ✔ | ✗ | ✗ | ✗ | ✗ | ✗ | ✗ |
| Criterion 2 | Water system | Water elements | relationship with water | water cross | ✗ | ✗ | ✗ | ✗ | ✔ | ✗ | ✗ | ✗ | ✗ | ✗ |
| | | | | water surround | ✗ | ✗ | ✗ | ✗ | ✗ | ✔ | ✗ | ✗ | ✗ | ✗ |
| | | | | water face to village | ✗ | ✗ | ✗ | ✗ | ✗ | ✗ | ✔ | ✗ | ✔ | ✗ |
| | | | | water scatter | ✗ | ✗ | ✗ | ✗ | ✗ | ✗ | ✗ | ✔ | ✗ | ✔ |
| Criterion 3 | Road system | Road elements | relationship with road | Built depend on mountain | ✔ | ✔ | ✔ | ✔ | ✗ | ✗ | ✗ | ✗ | ✗ | ✗ |
| | | | | Built depend on river | ✗ | ✗ | ✗ | ✗ | ✔ | ✔ | ✔ | ✔ | ✗ | ✗ |
| | | | | Built depend on river with mountain | ✗ | ✗ | ✗ | ✗ | ✗ | ✗ | ✗ | ✗ | ✔ | ✔ |
| Criterion X (X ≥ 3) | Other qualitative factors | ——— | | | | | | | | | | | | |

**Table A3.** The Criterion Framework for Traditional Village Spatial Organization Pedigree: Edited by Authors.

| Level | Macro Level | Meso Level | | | | | | | | |
|---|---|---|---|---|---|---|---|---|---|---|
| | | Appendix A-2 The Criterion Framework for Traditional Village Spatial Organization Pedigree | | | | | | | Net Radial Pedigree | Random Scatter Pedigree |
| Research Object | Influencing Factors | Prototype | | | Centripetal Aggregation Pedigree | | | | | |
| | | | | Sub-Types | Ordered Single-Center Aggregation | Ordered Multi-Center Aggregation | Disordered Single-Center Aggregation | Disordered Multi-Center Aggregation | | |
| Criterion 1 | Elevation | Terrain features | | plain | ✔ | ✔ | ✗ | ✗ | ✗ | ✗ |
| | | | | non-plain | ✗ | ✗ | ✔ | ✔ | ✔ | ✔ |
| | | Village architecture group | Number of village architecture group | 0 | ✗ | ✗ | ✗ | ✗ | ✗ | ✔ |
| | | | | >3 | ✗ | ✗ | ✗ | ✗ | ✔ | ✗ |
| | | | The number of architecture units in the group > 2 | | ✗ | ✗ | ✗ | ✔ | ✗ | ✗ |
| Criterion 2 | Water system | Water system style | Village axis | Show a straight axis that coincides with the street | ✔ | ✔ | ✗ | ✗ | ✗ | ✗ |
| | | | | Show a tortuous axis that does not necessarily coincide with the street | ✗ | ✗ | ✔ | ✔ | ✗ | ✔ |
| | | | | No axis | ✗ | ✗ | ✗ | ✗ | ✔ | ✗ |
| Criterion 3 | Road system | Road system style | Public space elements coincide with road nodes | | ✔ | ✔ | ✔ | ✔ | ✗ | ✗ |
| | | | Shape | straight | ✔ | ✔ | ✗ | ✗ | ✗ | ✗ |
| | | | | changeable | ✗ | ✗ | ✔ | ✔ | ✔ | ✔ |
| Criterion X (X ≥ 3) | Other qualitative factors | Public space elements are used for village's daily activities and gathering centers (including squares, ancestral halls, temples, ancient trees, ancient wells, etc.). | Number of public space elements | 0 | ✗ | ✗ | ✗ | ✗ | ✔ | ✗ |
| | | | | 1 | ✔ | ✗ | ✔ | ✗ | ✗ | ✔ |
| | | | | >1 | ✗ | ✔ | ✗ | ✔ | ✗ | ✔ |
| | | | Clusters of buildings are formed around the elements | | ✔ | ✔ | ✔ | ✔ | ✗ | ✗ |
| | | | The number of users > 50% of the total number of villagers | | ✔ | ✔ | ✔ | ✔ | ✗ | ✗ |
| | | | 15 min walking service radius | | ✔ | ✔ | ✔ | ✗ | ✗ | ✗ |

**Table A4.** The Criterion Framework for Traditional Village Spatial Morphological Pedigree: Edited by Authors.

| Macro Level | Meso Level | | | | | | | | | | |
|---|---|---|---|---|---|---|---|---|---|---|---|
| | **Appendix A-3 The Criterion Framework for Traditional Village Spatial Morphological Pedigree** | | | | | | | | | | |
| **Influencing Factors** | **Prototype** | | | **Regular Form** | | | **Irregular Form** | | | **Artistic Image Form** | |
| | **Sub-Types** | | | **Polygon Form** | **Round/Oval Form** | **Sector Form** | **Freestyle Form** | **Scatter-Point Form** | **Banding Form** | **Object Imaginative Art** | **Animal Imaginative Art** |
| Elevation | Village outline boundary style | Geometric shape | Polygon | ✔ | ✗ | ✗ | ✗ | ✗ | ✗ | ✗ | ✗ |
| | | | Round/oval | ✗ | ✔ | ✗ | ✗ | ✗ | ✗ | ✗ | ✗ |
| | | | Sector | ✗ | ✗ | ✔ | ✗ | ✗ | ✗ | ✗ | ✗ |
| | | Number of village space unit | 1 | ✔ | ✔ | ✔ | ✔ | ✗ | ✔ | ✔ | ✗ |
| | | | >2 | ✗ | ✗ | ✗ | ✗ | ✔ | ✗ | ✗ | ✔ |
| Water system | Water system style | Village axis | Show a straight axis that coincides with the water system | ✔ | ✗ | ✔ | ✗ | ✗ | ✔ | ✗ | ✗ |
| | | | Show a tortuous axis that does not necessarily coincide with the water system | ✗ | ✔ | ✗ | ✔ | ✗ | ✗ | ✔ | ✔ |
| | | | No axis | ✗ | ✗ | ✗ | ✗ | ✔ | ✗ | ✗ | ✗ |
| Road system | Village road style | Straight | | ✔ | ✗ | ✗ | ✗ | ✗ | ✔ | ✗ | ✗ |
| | | Sector angle | | ✗ | ✗ | ✔ | ✗ | ✗ | ✗ | ✗ | ✗ |
| | | Curved like S | | ✗ | ✗ | ✗ | ✗ | ✗ | ✔ | ✗ | ✗ |
| | | Cross like X | | ✗ | ✗ | ✗ | ✗ | ✗ | ✔ | ✗ | ✗ |
| | | Changeable like the net | | ✗ | ✗ | ✗ | ✔ | ✔ | ✗ | ✔ | ✔ |
| Other qualitative factors | Feng shui planning principles | | | ✔ | ✔ | ✔ | ✔ | ✗ | ✗ | ✔ | ✔ |
| | Local art worship | | | ✗ | ✗ | ✗ | ✗ | ✗ | ✗ | ✔ | ✔ |
| | Artistic spatial form interpretation of official documents | | | ✔ | ✔ | ✔ | ✗ | ✗ | ✗ | ✔ | ✔ |

**Table A5.** The Criterion Framework for Traditional Village Architectural Form Pedigree: Edited by Authors.

| Level | Research Object | Prototype | Sub-types | Criterion 1 — Elevation — Area | | | | | Criterion 2 — Water System — relationship with water | | | Criterion 3 — Road System — relationship with road | | number of courtyards | | | Criterion 4 — Other Qualitative Factors — Architecture elements | | | | | |
|---|---|---|---|---|---|---|---|---|---|---|---|---|---|---|---|---|---|---|---|---|---|---|
| Macro Level | Influencing Factors | | | Shanghai | Jiangsu | Anhui | Jiangxi | Zhejiang | near | front | cross | beside | slope | 1 | 2 | ≥3 | garden | western elements | Hui-style elements | bark | Multi-way and wall | corridors |
| Micro level | Appendix A-4 The Criterion Framework for Traditional Village Architectural Form Pedigree — Shanghai residence | | Near-water residence | ✔ | ✗ | ✗ | ✗ | ✗ | ✔ | ✗ | ✗ | ✗ | ✗ | ✗ | ✗ | ✗ | ✗ | ✗ | ✗ | ✗ | ✗ | ✗ |
| | | | Street-beside residence | ✔ | ✗ | ✗ | ✗ | ✗ | ✗ | ✗ | ✗ | ✔ | ✗ | ✗ | ✗ | ✗ | ✗ | ✗ | ✗ | ✗ | ✗ | ✗ |
| | | | Courtyard residence | ✔ | ✗ | ✗ | ✗ | ✗ | ✗ | ✗ | ✗ | ✗ | ✗ | ✔ | ✗ | ✗ | ✗ | ✗ | ✗ | ✗ | ✗ | ✗ |
| | | | Chinese and Western combined residence | ✔ | ✗ | ✗ | ✗ | ✗ | ✗ | ✗ | ✗ | ✗ | ✗ | ✗ | ✗ | ✗ | ✗ | ✔ | ✗ | ✗ | ✗ | ✗ |
| | | | Modern garden house | ✔ | ✗ | ✗ | ✗ | ✗ | ✗ | ✗ | ✗ | ✗ | ✗ | ✗ | ✗ | ✗ | ✔ | ✗ | ✗ | ✗ | ✗ | ✗ |
| | | Southern Jiangsu residence | Waterfront residence | ✗ | ✔ | ✗ | ✗ | ✗ | ✗ | ✔ | ✗ | ✗ | ✗ | ✗ | ✗ | ✗ | ✗ | ✗ | ✗ | ✗ | ✗ | ✗ |
| | | | Single-family residence | ✗ | ✔ | ✗ | ✗ | ✗ | ✗ | ✗ | ✗ | ✔ | ✗ | ✗ | ✗ | ✗ | ✗ | ✗ | ✗ | ✗ | ✗ | ✗ |
| | | | Cross-water residence | ✗ | ✔ | ✗ | ✗ | ✗ | ✗ | ✗ | ✔ | ✗ | ✗ | ✗ | ✗ | ✗ | ✗ | ✗ | ✗ | ✗ | ✗ | ✗ |
| | | | Multi-way multi-entry residence | ✗ | ✔ | ✗ | ✗ | ✗ | ✗ | ✗ | ✗ | ✗ | ✗ | ✗ | ✗ | ✗ | ✗ | ✗ | ✗ | ✗ | ✔ | ✗ |
| | | | Garden residence | ✗ | ✔ | ✗ | ✗ | ✗ | ✗ | ✗ | ✗ | ✗ | ✗ | ✗ | ✗ | ✗ | ✔ | ✗ | ✗ | ✗ | ✗ | ✗ |
| | | | Chinese and Western combined residence | ✗ | ✔ | ✗ | ✗ | ✗ | ✗ | ✗ | ✗ | ✗ | ✗ | ✗ | ✗ | ✗ | ✗ | ✔ | ✗ | ✗ | ✗ | ✗ |

**Table A5.** *Cont.*

| Level | Research Object | Prototype | Sub-types | Criterion 1 — Elevation (Area) | | | | | Criterion 2 — Water System (relationship with water) | | | Criterion 3 — Road System (relationship with road) | | number of courtyards | | | Criterion 4 — Other Qualitative Factors (Architecture elements) | | | | | |
|---|---|---|---|---|---|---|---|---|---|---|---|---|---|---|---|---|---|---|---|---|---|---|
| Macro Level | Influencing Factors | | | Shanghai | Jiangsu | Anhui | Jiangxi | Zhejiang | near | front | cross | beside | slope | 1 | 2 | ≥3 | garden | western elements | Hui-style elements | bark | Multi-way and wall | corridors |
| Micro level | Appendix A-4 The Criterion Framework for Traditional Village Architectural Form Pedigree | Southern Anhui residence Northeast Jiangxi | Bark residence | ✗ | ✗ | ✔ | ✗ | ✗ | ✗ | ✗ | ✗ | ✗ | ✗ | ✗ | ✗ | ✗ | ✗ | ✗ | ✗ | ✔ | ✗ | ✗ |
| | | | Stone residence | ✗ | ✗ | v | ✗ | ✗ | ✗ | ✗ | ✗ | ✗ | ✔ | ✗ | ✗ | ✗ | ✗ | ✗ | ✗ | ✗ | ✗ | ✗ |
| | | | Earth wall residence | ✗ | ✗ | ✔ | ✗ | ✗ | ✗ | ✗ | ✗ | ✗ | ✗ | ✗ | ✗ | ✗ | ✗ | ✗ | ✗ | ✗ | ✔ | ✗ |
| | | | Hui-style residence | ✗ | ✗ | ✔ | ✗ | ✗ | ✗ | ✗ | ✗ | ✗ | ✗ | ✗ | ✗ | ✗ | ✗ | ✗ | ✔ | ✗ | ✗ | ✗ |
| | | | Waterfront residence | ✗ | ✗ | ✗ | ✔ | ✗ | ✗ | ✔ | ✗ | ✗ | ✗ | ✗ | ✗ | ✗ | ✗ | ✗ | ✗ | ✗ | ✗ | ✗ |
| | | | Slope residence | ✗ | ✗ | ✗ | ✔ | ✗ | ✗ | ✗ | ✗ | ✗ | ✔ | ✗ | ✗ | ✗ | ✗ | ✗ | ✗ | ✗ | ✗ | ✗ |
| | | | Courtyard residence | ✗ | ✗ | ✗ | ✔ | ✗ | ✗ | ✗ | ✗ | ✗ | ✗ | ✔ | ✗ | ✗ | ✗ | ✗ | ✗ | ✗ | ✗ | ✗ |
| | | | Hui-style residence | ✗ | ✗ | ✗ | ✔ | ✗ | ✗ | ✗ | ✗ | ✗ | ✗ | ✗ | ✗ | ✗ | ✗ | ✗ | ✔ | ✗ | ✗ | ✗ |
| | | | Horse head wall | ✗ | ✗ | ✗ | ✔ | ✗ | ✗ | ✗ | ✗ | ✗ | ✗ | ✗ | ✗ | ✗ | ✗ | ✗ | ✗ | ✗ | ✔ | ✗ |
| | | Northeast Zhejiang residence | Water town residence | ✗ | ✗ | ✗ | ✗ | ✔ | ✔ | ✗ | ✗ | ✗ | ✗ | ✗ | ✗ | ✗ | ✗ | ✗ | ✗ | ✗ | ✗ | ✗ |
| | | | Hangzhou style residence | ✗ | ✗ | ✗ | ✗ | ✔ | ✗ | ✗ | ✗ | ✗ | ✗ | ✗ | ✗ | ✗ | ✗ | ✗ | ✗ | ✗ | ✗ | ✔ |
| | | | Garden style residence | ✗ | ✗ | ✗ | ✗ | ✔ | ✗ | ✗ | ✗ | ✗ | ✗ | ✔ | ✗ | ✗ | ✗ | ✗ | ✗ | ✗ | ✗ | ✗ |
| | | | Three pushes nine Mingtang | ✗ | ✗ | ✗ | ✗ | ✔ | ✗ | ✗ | ✗ | ✗ | ✗ | ✗ | ✔ | ✗ | ✗ | ✗ | ✗ | ✗ | ✔ | ✗ |
| | | | New big wall door | ✗ | ✗ | ✗ | ✗ | ✔ | ✗ | ✗ | ✗ | ✗ | ✗ | ✗ | ✗ | | ✗ | ✔ | ✗ | ✗ | ✔ | ✗ |
| | | | Taimen | ✗ | ✗ | ✗ | ✗ | ✔ | ✗ | ✗ | ✗ | ✗ | ✗ | ✗ | ✗ | ✔ | ✗ | ✗ | ✗ | ✗ | | ✗ |
| | | | Large wall door | ✗ | ✗ | ✗ | ✗ | ✔ | ✗ | ✗ | ✗ | ✗ | ✗ | ✔ | ✗ | | ✗ | ✗ | ✗ | ✗ | ✔ | ✗ |
| | | | Inter-alley pavilion | ✗ | ✗ | ✗ | ✗ | ✔ | ✗ | ✗ | ✗ | ✗ | ✗ | ✗ | ✗ | ✗ | ✗ | ✗ | ✗ | ✗ | ✔ | ✗ |
| | | | Walk the horse | ✗ | ✗ | ✗ | ✗ | ✔ | ✗ | ✗ | ✗ | ✗ | ✗ | ✗ | ✗ | ✗ | ✗ | ✗ | ✔ | ✗ | ✗ | ✗ |
| | | | Eighteen Floor | ✗ | ✗ | ✗ | ✗ | ✔ | ✗ | ✗ | ✗ | ✗ | ✗ | ✗ | ✔ | | ✗ | ✗ | ✗ | ✗ | ✗ | ✗ |

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
