# Peer review of "The Spatial Pedigree in Traditional Villages under the Perspective of Urban Regeneration—Taking 728 Villages in Jiangnan Region, China as Cases"

_land, doi:10.3390/land11091561_

Round 1
Reviewer 1 Report
Please find the comments in the attached file

Reviewer 2 Report
A very interesting and well-written paper
It would be interesting to compare the architectural patterns present in contemporary villages with other parts of the world (maybe in the literature overview section, or generaly). What can be done to preserve the traditional rural architectural tissue?
Reviewer 3 Report
There is a very valuable, well-designed and worth reading paper. The authors did a big job to prove their competencies and good quality workshop. The article includes an excellent method to find out and use the spatial code of the place, providing a helpful procedure to protect the essential landscape values.
I think adding a baseline of the method in the abstract will be good, making the article easier to find for more readers.
Round 2
Reviewer 1 Report
Please find the comments in the attached file
